# LMO-DP: Accurately Fine-Tuning Language Models with Stronger Differential Privacy

## Abstract

Differentially Private Stochastic Gradient Descent (DP-SGD) and its variants have been proposed to ensure rigorous privacy for fine-tuning large-scale pre-trained language models. State-of-the-art (SOTA) DP-SGD methods rely heavily on the Gaussian mechanism, which may overly perturb the gradients and degrade the fine-tuning accuracy, especially in stronger privacy regimes (e.g., the total privacy budget $\epsilon < 3$).[1] To address such limitations, we propose a novel Language Model-based Optimal Differential Privacy (LMO-DP) framework, which takes the first step to enable the tight composition of a sub-optimal DP mechanism (non-Gaussian) for accurately fine-tuning language models, even in stronger privacy regimes (e.g., $0.5 \leq \epsilon < 3$). Furthermore, LMO-DP efficiently approximates the sub-optimal DP and fast convergence, compared to the SOTA methods. For instance, fine-tuning RoBERTa-large (with 300M parameters) on the SST-2 dataset can achieve the 92.20% accuracy (given the total privacy budgets $\epsilon = 0.3$ and $\delta = 0$), compared with the $\sim$50% accuracy of most SOTA methods. We also draw similar findings on text generation tasks while privately fine-tuning GPT-2.

## 1 Introduction

Recently large language models (LLMs) have achieved breakthrough success by effectively processing and encoding large text data from extremely large-scale training datasets. For instance, BERT (Liu et al. (2019)) and GPT families (Yu et al. (2021)) have demonstrated state-of-the-art (SOTA) accuracy and enhanced performance in most of the learning tasks. In addition, such (open-sourced) language models are pre-trained on extremely large and generic datasets, and then fine-tuned to accurately support a wide variety of downstream tasks using a relatively smaller dataset in the task domain, e.g., sentence classification (Liu et al. (2019)), text generation (Novikova et al. (2017)), and code generation (Wang et al. (2018)).

On the other hand, deep learning models have been proven to be vulnerable to privacy threats during training (Abadi et al. (2016); Shokri et al. (2017); Hayes et al. (2017)). Similar to models in other domains, training and/or fine-tuning language models would potentially leak sensitive information. Notice that, since pre-trained datasets and models have been published, they have already rendered privacy leakage from the open-sourced datasets and models. Thus, it is desirable to privately fine-tune language models by protecting the sensitive information in the new dataset used for fine-tuning (Li et al. (2022); Yu et al. (2021); Bu et al. (2022b); He et al. (2022); Bu et al. (2023)).

To mitigate privacy risks in deep learning training and fine-tuning, differential privacy (DP) (Dwork (2006)) has been widely recognized as the de facto rigorous privacy model where adding or removing any data sample, record, or user in the (training) data would not cause significant leakage. In particular, the well-known Differentially Private Stochastic Gradient Descent (DP-SGD) (Abadi et al. (2016)) method tightly balances the privacy and utility within the training, leveraging the privacy budget parameters $\epsilon$ and $\delta$ in Gaussian mechanism. It achieves this by constraining the influence of individual examples through gradient clipping and adding Gaussian noise to the gradients within each batch, providing a DP guarantee for neural network training (Abadi et al. (2016)).

To our best knowledge, the SOTA methods (which are DP-SGD variants) (Li et al. (2022); He et al. (2022); Panda et al. (2023); Yu et al. (2021); Bu et al. (2022b; 2023)) for privately fine-tuning lan-

---

[1]Most SOTA methods showed high accuracy in case of relatively weaker DP, e.g., $\epsilon \geq 3$, but not small $\epsilon$.

guage models mainly focus on optimizing the gradient clipping mechanism to enhance utility and/or improve system performance (e.g., reducing memory) while maintaining privacy. Furthermore, such DP-SGD based methods rely heavily on the Gaussian mechanism since its key component – moment accountant (MA) leverages the properties of Gaussian noise to accumulate the overall privacy budget via a tight DP composition. However, the privacy constraints imposed in DP-SGD, solely on the Gaussian noise, may still overly perturb the gradients and degrade the accuracy, especially when the total privacy budget is small, e.g., $0.5 \leq \epsilon < 3$ (stronger privacy regimes).

## 1.1 CHALLENGES AND CONTRIBUTIONS

To boost the tradeoff between privacy and accuracy for language models (LMs), we propose a novel Language Model-based Optimal Differential Privacy (LMO-DP) framework. To this end, we first generalize the moment accountant (MA) in DP-SGD to universally support a tight DP composition based on randomization mecha-

Table 1: Sentiment classification (RoBERTa-large on SST-2). "-" means not able to obtain (acc.) or not compared. Fast convergence is based on empirical observations.

| Methods | Acc. ($\epsilon = 0.3$) | Fast Conv. | Memory | DP | Noise |
|---|---|---|---|---|---|
| Li et al. (2022) | 49.43% | × | low | $(\epsilon, \delta)$-DP | Gaussian |
| Yu et al. (2021) | - | - | low | $(\epsilon, \delta)$-DP | Gaussian |
| Bu et al. (2022b) | 50.52% | - | low | $(\epsilon, \delta)$-DP | Gaussian |
| He et al. (2022) | - | - | low | $(\epsilon, \delta)$-DP | Gaussian |
| Bu et al. (2023) | 50.52% | - | low | $(\epsilon, \delta)$-DP | Gaussian |
| **LMO-DP (Ours)** | 92.20% | ✓ | low | $\epsilon$-DP | sub-optimal |

nisms (other than Gaussian). Then, LMO-DP efficiently approximates a sub-optimal DP (Mohammady et al. (2020)) for the LM fine-tuning. It is worth noting that LMO-DP can also strictly satisfy $\epsilon$-DP, rather than $(\epsilon, \delta)$-DP, relaxed by the Gaussian mechanism. Table 1 summarizes the comparison between LMO-DP with SOTA methods in an example setting (sentiment classification tasks on the SST-2 dataset with the RoBERTa-large model). Specifically, we discuss the new challenges and major contributions of LMO-DP as below:

**(1) Tight Composition for Sub-optimal DP**. Rényi Differential Privacy (Rényi-DP) (Mironov (2017)) provides a tight privacy composition of MA on Gaussian mechanism (Wang et al. (2019b)). Hence, Rényi-DP can be leveraged to generalize MA to other DP mechanisms, such as Laplace (Dwork et al. (2006)), Staircase (Geng & Viswanath (2014)), Matrix-Variate Gaussian (MVG) (Chanyaswad et al. (2018)), and mixture distribution for optimal DP (Mohammady et al. (2020)). To improve the privacy/accuracy tradeoff, LMO-DP will extend the MA via Rényi-DP to ensure a tight composition of the mixture distribution for approximating a sub-optimal DP (Mohammady et al. (2020)). Its Rényi-DP guarantee will be derived for calculating the total privacy loss.

**(2) Efficiently Approximating Sub-optimal DP and Fast Convergence**. Due to the high demand on efficiency for LM training and fine-tuning, it is not practical to directly apply end-to-end optimal DP (Mohammady et al. (2020)) during each step of fine-tuning. Alternatively, LMO-DP efficiently approximates the sub-optimal randomization mechanism for LM fine-tuning. To address this major challenge, we establish a meticulously defined search space of Probability Density Functions (PDFs) while upholding a universal privacy guarantee. This search space closely approximates the entire spectrum of conceivable PDFs. Then, we construct a framework that empowers optimization within this space, ultimately leading to notable advancements in both accuracy and convergence for LMs.[2]

Specifically, we introduce a subspace within the PDF space by incorporating randomization into the scale parameter of Laplace noise. By instantiating this randomization as a linear combination of *Gamma*, *Exponential*, and *Uniform* distributions, we approximate the entirety of the search space of PDFs. This innovative approach enables us to formulate privacy and utility within a unified framework. One of the most remarkable outcomes of LMO-DP is the feasibility of achieving high levels of accuracy even under *very strong DP guarantees*, such as $\epsilon = 0.3$. Our experiments have revealed $90\%+$ accuracy, effectively highlighting the potential of LMO-DP in effectively balancing privacy and utility. Meanwhile, We also found empirically LMO-DP achieves superior convergence rates in a diverse range of LM tasks (e.g., sentiment classification, table-to-text generation).

**(3) Achieving Low Memory via Universal Integration with Orthnogal Methods**. Since LMO-DP mainly optimizes the randomization for DP mechanisms offline (pre-processing), it is orthogonal to existing DP-SGD based fine-tuning methods for LMs (e.g., Bu et al. (2023)). Thus, LMO-DP can inherit all the benefits of existing methods via integration with them, e.g., memory reduction

---

[2]The approximated optimization (for deriving the sub-optimal PDF) will be executed as a pre-processing procedure before the fine-tuning. Then, LMO-DP would only incur minor extra runtime during fine-tuning.

via Ghost Clipping (Li et al. (2022)). For instance, our LMO-DP implementation has two modes, including the one integrated with the Ghost Clipping, which requires low memory (see Table 1).

In summary, to our best knowledge, we are the first to fine-tune LMs (including the GPT-2 (Li et al. (2022))) with an approximated sub-optimal DP (which demonstrates significantly higher accuracy for stronger privacy regimes and faster convergence), and also the first to design non-Gaussian DP mechanism for strictly ensuring $\epsilon$-DP where $\delta = 0$. We focus on fine-tuning in this work since this is popular in practice for language models (LMs) and it can also be readily extended to training from scratch and other applications.

## 2 RELATED WORK

**Parameter-based LM Fine-tuning with DP**. DP-SGD (Abadi et al. (2016)) was initially devised for private neural network training. However, a significant challenge with traditional DP-SGD lies in compromised performance and the substantial time and memory overhead by private training. Researchers are actively addressing it by reducing training costs and improving performances for language models through more efficient parameter tuning. For instance, Yu et al. (2021) enhanced DP-SGD by exploring parameter-efficient tuning methods that focus on training only a fraction of the model parameters, resulting in improved utility, privacy, and reduced overheads. Another notable advancement comes from Bu et al. (2022b), introducing a model-agnostic DP bias-term fine-tuning (DP-BiTFiT) framework. It prioritizes optimizing the bias rather than model weights, and achieves efficiency by activating only the backward hook in PyTorch, thus saving time and space.

**Clipping-based LM Fine-tuning with DP**. Additionally, researchers have introduced novel clipping methods to reduce computational time and memory requirements for large language models. Li et al. (2022); Bu et al. (2022a) introduced the ghost clipping method, significantly reducing memory usage during training and enhancing performance in text classification and generation tasks. He et al. (2022) proposed adaptive group-wise clipping, encompassing per-layer and per-device clipping techniques, suitable for deployment on multiple accelerators. Bu et al. (2022a) proposed a mixed ghost clipping method on convolutional layers, that significantly eases the private training in terms of both time and space while maintaining the accuracy. Bu et al. (2023) introduces a novel book-keeping (BK) technique that enhances the computational efficiency by eliminating the need for a second back-propagation step in GhostClip (Bu et al. (2022b)), while preserving the same accuracy.

**Other Work**. Panda et al. (2023) adopted the Gaussian mechanism for private prediction instead of private training. This eliminates the need for hyperparameter tuning and ensures efficiency.

## 3 PRELIMINARIES

### 3.1 DIFFERENTIAL PRIVACY AND DP-SGD

We first define differential privacy (see Appendix A.1 for the definition of probability space):

**Definition 1 ($(\epsilon, \delta)$-Differential Privacy)** *A randomization mechanism $\mathcal{M}$ satisfies $(\epsilon, \delta)$-differential privacy if, for any two adjacent datasets $D$ and $D'$, and for any subset of outputs $S \subseteq range(\mathcal{M})$, $\Pr[\mathcal{M}(D) \in S] \leq e^\epsilon \Pr[\mathcal{M}(D') \in S] + \delta$.*

DP-SGD (Abadi et al. (2016)) was introduced to guarantee differential privacy during deep learning training, incorporating the Gaussian mechanism. It works by first clipping gradients $\mathbf{g}_t(x_i)$ using a threshold $C$ for $\ell_2$-sensitivity $\overline{\mathbf{g}}_t(x_i) = \mathbf{g}_t(x_i) \max\left(1, \frac{\|\mathbf{g}_t(x_i)\|_2}{C}\right)$. Gaussian noise $\mathcal{N}(0, C^2\sigma^2\mathbf{I})$ is then added to these clipped gradients within each batch $\tilde{\mathbf{g}}_t = \frac{1}{L}\left(\sum_i \overline{\mathbf{g}}_t(x_i) + \mathcal{N}(0, C^2\sigma^2\mathbf{I})\right)$.

### 3.2 RÉNYI-DP MOMENT ACCOUNTANT FOR DP-SGD

Rényi-DP offers a more flexible privacy measure than traditional DP, characterized by Rényi divergences. It measures the closeness between probability distributions of adjacent datasets that differ by a single data sample. Specifically, $\mathcal{M}$ is $(\alpha, \epsilon_\alpha)$-Rényi-DP with order $\alpha \in (1, \infty)$ if for all adja-

cent datasets $D$ and $D'$; $D_\alpha(\mathcal{M}(D)\|\mathcal{M}(D')) \leq \epsilon_\alpha$, where $D_\alpha$ is defined per the expectations over $\theta \sim \mathcal{M}(D')$. As $\alpha \to \infty$, Rényi-DP reduces to traditional $(\epsilon, 0)$-DP.

The Rényi-DP Moment Accountant is a tool used to analyze privacy guarantees in DP-SGD per Rényi-DP. It calculates the cumulative Rényi-DP privacy guarantee based on a sequence of Rényi-DP mechanisms $\mathcal{M}_1, \ldots, \mathcal{M}_t$ with associated privacy budgets $\epsilon_1, \ldots, \epsilon_t$ and order $\alpha \in (1, \infty)$:

$$\epsilon_\alpha = \sum_{i=1}^{t} \left[ \frac{1}{\alpha - 1} \cdot \log \left( \mathbb{E}_{\theta \sim \mathcal{M}_i} \left[ \left( \frac{\mathcal{M}_i(\theta)}{\mathcal{M}_{i-1}(\theta)} \right)^\alpha \right] \right) \right] \tag{1}$$

Then, we can derive tight privacy bounds for DP-SGD, ensuring specified privacy (e.g., $\epsilon$) while controlling the privacy failure probability (e.g., $\delta$) (Wang et al. (2019a)). Specifically, for a small $\delta$ (e.g., $10^{-5}$), the total privacy loss $\epsilon$ is $\epsilon(\delta) = \min_{\alpha > 1} \left\{ \log(\log \frac{1}{\delta} + \epsilon_\alpha)/\alpha \right\}$ which is represented by the privacy curve in Gopi et al. (2021).

### 3.3 NOISE OPTIMIZATION IN DP-SGD FOR LANGUAGE MODELS (LMs)

We next formalize key concepts, including language model (LM) geometry and the formulation of Rényi-DP for PDFs $\mathbb{P}_{i/o}^D$ and $\mathbb{P}_{i/o}^{D'}$. Our objective is to define an optimization problem for the cross-entropy of the LM geometry, subject to an $(\alpha, \epsilon_\alpha)$-Rényi-DP constraint.

**LM Geometry**. The "Geometry of LMs" is denoted as $(\Omega, \mathcal{F})$, where $\Omega$ is the LM's parameter space, and $\mathcal{F}$ includes functions mapping model configurations within $\Omega$ to specific characteristics.

Specifically, LM Geometry is defined as $\mathbb{P}_{i/o}$, where $i$ and $o$ denote the input and output of the training task, respectively. $\mathbb{P}_{i/o}$ characterizes the probabilistic transition from $i$ to $o$, enabling us to quantify and analyze LM geometric properties.

**Global Optimization**. Given the LM model geometry for any pair of adjacent datasets $D$ and $D'$, we formulate the following optimization problem over all choice of perturbed LM geometry $\tilde{\mathbb{P}}_{i/o}^D$:

$$\min_{\tilde{\mathbb{P}}_{i/o}^D : \mathcal{F} \to [0,1]} - \log \left( \frac{1}{N} \sum_{i=1}^{N} \tilde{\mathbb{P}}_{i/o^c}^D \right) \quad \text{(Cross-entropy loss: penalizing smaller probabilities for correct class } c\text{)}$$

$$\text{s.t.} \quad \frac{1}{\alpha - 1} \sup_{\forall \{D, D'\} \in \mathcal{D}} \log \left( \frac{1}{N} \sum_{i=1}^{N} \sum_{o=1}^{C} \left( \tilde{\mathbb{P}}_{i/o}^D \right)^\alpha \cdot \left( \tilde{\mathbb{P}}_{i/o}^{D'} \right)^{1-\alpha} \right) \leq \epsilon_\alpha \tag{2}$$

where the privacy bound $\epsilon$ is defined as $\epsilon(\delta) = \min_{\alpha > 1} \left\{ \log \left( \frac{\log \frac{1}{\delta} + \epsilon_\alpha}{\alpha} \cdot \right) \right\}$, $C$ is the number of output classes, $\alpha$ is the order of Rényi-DP, and $D = \{x_j\}_{j=1}^{N}$ is the training dataset.

The cross-entropy loss penalizes perturbed geometries with smaller probabilities for the correct labeling. This choice of loss function is particularly suitable for sequencing tasks and aligns with the probabilistic interpretation commonly used in training LMs which accounts for allowing the LM to learn to generate sequences that closely resemble the true data distribution. In addition, each constraint is defined by the Rényi-DP of the LM geometry. The range of $\alpha$ can be specified empirically, e.g., 2,...,128.[3] These constraints are derived and applied using the moment accountant, which allows for tight control and accounting of the privacy budget across multiple rounds of fine-tuning, ensuring that the LM's privacy guarantees are upheld overall in our paper. Notice that, many recent works (Zhu et al. (2022); Koskela et al. (2023)) provide tighter privacy accounting for DP, which could also be applied to formulate similar constraints. Then, we can replace the Rényi-DP with those new accountants with tighter bounds, though different solvers will be desirable.

## 4 LMO-DP WITH SUB-OPTIMAL RANDOMIZATION

In this section, we transform the global optimization in Eq. (2) into a sub-optimal but more manageable problem. Specifically, we will demonstrate that for the class of cross-entropy loss functions, we

---

[3]Expanding the upper bound of $\alpha$ can facilitate the computation of Rényi privacy. Nevertheless, opting for larger values may result in moments too tiny for ineffective processing of floating point numbers.

can approximate the optimal solution by optimizing a series of Rényi-DP problems, effectively eliminating the need for the original objective function in Eq. (2). We will next introduce our proposed methodology to efficiently find the sub-optimal randomized LM geometry.

## 4.1 SUB-OPTIMAL REDUCTION

We first give a sub-optimal reduction to Eq. (2). We argue that fixing $\delta$, and assuming a monotonic relationship between $\epsilon$ and the accuracy of an $(\epsilon, \delta)$-DP LM geometry $\tilde{\mathbb{P}}_{i/o}^D$ can be deemed as a good candidate to optimize Eq. (2) if the maximum $\alpha$-th Rényi-DP constraint (for all $\alpha > 1$) is satisfied.

**Theorem 4.1 (Proven in Appendix B.1)** *Given DP parameters $(\epsilon, \delta)$, consider the optimal $(\epsilon, \delta)$-DP LM geometry in Eq. (2), denoted as $X^D$ where $D$ is any dataset in the domain ($D'$ is its any adjacent dataset). Let $\epsilon(\delta)$ be the privacy curve (Gopi et al. (2021)) of $X^D$, and $\epsilon_\alpha$ denote its $\alpha$-order RDP moment. Given an $(\epsilon, \delta)$-DP LM geometry $\tilde{\mathbb{P}}_{i/o}^D$, we define the optimization problem:*

$$\min_{\tilde{\mathbb{P}}_{i/o}^D : \mathcal{F} \to [0,1], \forall \alpha > 1} \left\| \frac{1}{N} \log \left( \sum_{i=1}^N \sum_{o=1}^C \left( \tilde{\mathbb{P}}_{i/o}^D \right)^\alpha \cdot \left( \tilde{\mathbb{P}}_{i/o}^{D'} \right)^{1-\alpha} \right) + \epsilon_\alpha \right\|. \quad (3)$$

*Then, $\tilde{\mathbb{P}}_{i/o}^D$ sub-optimally solves the global optimization problem in Eq. (2).*

**Proof Sketch**. First, we establish a critical criterion: for all values of $\alpha$ and for all possible selections of $\tilde{\mathbb{P}}_{i/o}^D$ from the set $\mathcal{F}$, these geometries should minimize the distance between the Rényi-DP of $\tilde{\mathbb{P}}i/o^D$ ($\epsilon^{\tilde{\mathbb{P}}i/o^D}$) and $\epsilon_\alpha$. To achieve that, we formulate a series of optimization problems, which can be considered as a series of sieves filtering out geometries that do not perform well in tracking any of $\alpha > 1$ moments of the $\epsilon(\delta)$. Next, we show that such a "golden" geometry $\tilde{\mathbb{P}}i/o^D$, which is a tight approximation of $\epsilon(\delta)$, can lead to a sub-optimal solution for problem (2). The key tool for this claim is the Abel's inequality which is used to specify one lower-bound and one upper-bound for the Rényi-DP expression $\frac{1}{\alpha-1} \log \left( \frac{1}{N} \sum_{i=1}^N \sum_{o=1}^C \left( \tilde{\mathbb{P}}_{i/o}^D \right)^\alpha \cdot \left( \tilde{\mathbb{P}}_{i/o}^{D'} \right)^{1-\alpha} \right)$. In particular, we show that the lower-bound of the solution for this expression is proportional to the cross-entropy term $-\log \left( \frac{1}{N} \sum_{i=1}^N \tilde{\mathbb{P}}_{i/o^c}^D \right)$. This illustrates that $\tilde{\mathbb{P}}_{i/o}^D$ is a sub-optimal solution for problem (2).

Next, we will: (1) introduce a versatile search space for these probabilities in Section 4.2, and (2) present the LMO-DP algorithm, which is tailored to identify $(\epsilon, \delta)$-DP configurations that provide the highest accuracy, particularly those with the lowest cross-entropy, in Section 4.3.

## 4.2 LMO SPACE: A COMPREHENSIVE SEARCH SPACE

To formulate an optimization problem over LM geometries, we introduce the "LMO Space", a versatile search space designed to encompass a broad spectrum of probability density functions (PDFs). The LMO Space is characterized by two essential properties: (1) comprehensiveness in covering all $\mathbb{P} \in \mathcal{F}$, and (2) a universal DP guarantee function for all $\mathbb{P} \in$ LMO, to support/facilitate solving the optimization problem (3). We define the LMO Space by randomizing the scale parameter ($b$) of the Laplace distribution according to an infinite number of positively supported PDFs. This space serves as the foundation for optimization (see Appendix C). In probability theory, quantifying the comprehensiveness of a subset of probability functions lacks a universally accepted measure. Thus, we empirically assess its comprehensiveness compared to a universally simulated space via a novel quantification test defined using several well-known metrics such as KL divergence, $\ell_2$ distance, and earth mover's distance (EMD). The quantification is detailed in Algorithm 2 in Appendix D.

Figure 5 in Appendix C.2 shows that the LMO Space aligns closely with the simulated space. This dual randomization process creates a space where elements are the moment generating functions (MGFs) of second-fold PDFs. This construction offers versatility by enabling linear combinations of various positively supported PDFs in the second-fold PDF, thanks to MGF composability (Appendix A.2). Moreover, the LMO Space provides a universal Rényi-DP guarantee (Appendix B.2):

$$\forall X = \text{Lap}(x) \in \text{LMO Space} : e_\alpha^\epsilon(x) \propto \mathcal{O}\left( \frac{dM}{dx} \right).$$

Section 4.3 will detail how to choose this set of independent random variables. While this search space may not encompass the entire space, we will show its sufficiency for near-optimal accuracy through numerical results (Figure 6 in Appendix C.2) and experiments in various learning settings (Section 5). Thus LMO is a comprehensive and universally defined set of DP LM geometries.

## 4.3 LMO-DP Framework

The Rényi-DP accountant optimization in the LMO-DP first derives the optimal geometry from problem (3) (see Algorithm 3 in Appendix D). The resulting set of optimal configurations will be subsequently applied in Algorithm 1. This optimization operates through a two-level approach, hinging on a pair of $\epsilon$ and $\delta$ values, which play a pivotal role in two distinct optimization stages.

First, utilizing Theorem 4.2, the Rényi-DP ($\alpha$) of LMO is computed. Subsequently, $\epsilon_{LMO}(\delta)$ is determined through the initial optimization $\min_{\alpha>1}\left\{\frac{\log(\log\frac{1}{\delta}+\epsilon_\alpha)}{\alpha}\right\}$. This initial optimization is carried out across a range of Rényi-DP orders where $2 \leq \alpha < \alpha_{\max}$ where $\alpha_{\max}$ is the given maximum order in RDP. Finally, the optimal selection of LMO geometry $\mathbb{P}$ and its corresponding parameters, including $\theta$ and other LMO parameters (weights of different randomization PDFs and PDF parameters), is determined via a second optimization problem. This secondary optimization, which is $\min_{\alpha>1}\left\{\log\left(\frac{\log\frac{1}{\delta}+\epsilon_\alpha}{\alpha}\cdot\right)\right\}$ inside the optimization problem (2), seeks to minimize the discrepancy between $\epsilon_{LMO}(\delta)$ and $\epsilon$.

Algorithm 1 provides insights into the procedures associated with the mixture of three PDFs. Specifically, the Laplace scale parameter in the LMO-DP mechanism is modeled as a random variable, following a distribution created by a mixture of the Gamma distribution ($Y_1 \sim \Gamma(k,\theta)$), the Exponential distribution ($Y_2 \sim Exp(\lambda)$), and the Uniform distribution ($Y_3 \sim U(a,b)$).

---

**Algorithm 1: LMO-DP**

**Input:** $A_1$: privacy budget $\{\epsilon,\delta\}$; $A_2$: hyperparameters in NLP tasks $\{\mathcal{D},\eta_t,T,B,C,\mathcal{L},\Theta_0\}$ – the training data $\mathcal{D}=\{x_i\}_{i=1}^N$, learning rate $\eta_t$, the number of steps $T$, batch size $B$, clipping threshold $C$, loss function $\mathcal{L}$ and model parameters $\Theta_t$; $A_3$: parameters related to noise computation – selected PDFs for mixture, the max order $\alpha_{max}$ during Rényi-DP accountant computation, and searched ranges of distribution parameters and their weights $\mathcal{S}$)

**Output:** $\Theta_T$

    /* Step 1: Acquiring the sub-optimal parameters $S_{opt}$     */

1 **Call Algorithm 3 (Rényi-DP Accountant Optimization):** $S_{\mathbf{opt}} = F_1(A_1, A_3)$

    /* Step 2: Fine-tuning on NLP tasks     */

2 **foreach** $t$ *in [1, T]* **do**

3      draw a batch $B_t$ via Poisson sampling

4      **foreach** $x_i \in B_t$ **do**

5          $g_t(x_i) = \bigtriangledown_{\Theta_t}\mathcal{L}(\Theta_t, x_i)$          // each sample's gradient $g_t(x_i)$ is computed

6      $\bar{\mathbf{g}}_t(x_i) \leftarrow \mathbf{g}_t(x_i) \cdot \min\left(1, \frac{C}{\|\mathbf{g}_t(x_i)\|_2}\right)$      // each sample's gradient $g_t(x_i)$ is clipped

        /* Gradient randomization with noise generated with $S_{\text{opt}}$     */

7      $\tilde{\mathbf{g}}_t \leftarrow \frac{1}{B}\left(\sum_{i=1}^N \bar{\mathbf{g}}_t(x_i) + \mathcal{M}(S_{opt})\right)$      // $\mathcal{M}(S_{opt})$ is the noise from LMO-DP mechanism

8      $\Theta_{t+1} \leftarrow \Theta_t - \eta_t\tilde{\mathbf{g}}_t$

9 **Return** $\Theta_T$

---

**Theorem 4.2 (Rényi-DP of LMO Geometries)** . *The Rényi-DP guarantee ($\alpha > 1$) of the mixture distribution in each step in the training process (see Appendix B.2) can be derived as:*

$$\epsilon_{LMO\text{-}DP} = \frac{1}{\alpha-1}\log\left\{\frac{\alpha}{2\alpha-1}\cdot\frac{1}{[1-a_1(\alpha-1)\theta]^k}\cdot\frac{1}{[1-a_2(\alpha-1)\lambda^{-1}]}\cdot\frac{e^{a_3(\alpha-1)b}-e^{a_3(\alpha-1)a}}{a_3(\alpha-1)(b-a)}\right.$$
$$\left.+\frac{\alpha-1}{2\alpha-1}\cdot\frac{1}{[1-a_1(-\alpha)\theta]^k}\cdot\frac{1}{[1-a_2(-\alpha)\lambda^{-1}]}\cdot\frac{e^{a_3(-\alpha)b}-e^{a_3(-\alpha)a}}{a_3(-\alpha)(b-a)}\right\} \quad (4)$$

*where $a_1(\alpha-1) < \frac{1}{\theta}, a_2(\alpha-1) < \lambda, b > a, k > 0, \theta > 0$, and $\lambda > 0$ (parameters for the Gamma, Exponential, and Uniform distributions).*

**LMO-DP Noise vs Gaussian Noise**. LMO-DP noise shows a significantly smaller amplitude as shown in Figure 1 (given the same DP guarantee and shown in logarithmic scale). The reduction rate of LMO-DP noise compared to Gaussian noise is as high 95.13% for $\epsilon = 0.3$, and then slightly reduced to 87.31% as $\epsilon$ increases to 3. This also shows that LMO-DP has suerpior performance for strong privacy guarantees (small $\epsilon$). In addition, Figure 6 in Appendix F.2 also provides visual evidence of the significant improvements achieved by the LMO-DP noise compared to Gaussian noise in various settings and scenarios.

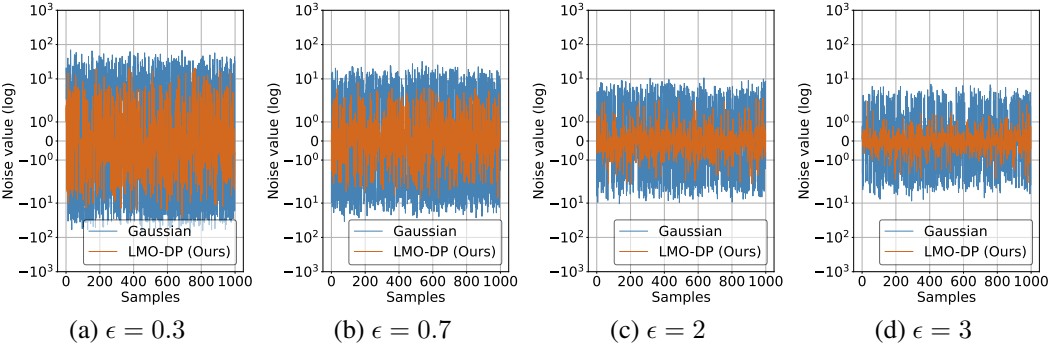

Figure 1: LMO-DP vs Gaussian. (a) average reduction rate $95.13\%$. (b) average reduction rate $92.19\%$. (c) average reduction rate $87.71\%$. (d) average reduction rate $87.31\%$. The results demonstrate that the LMO-DP noise significantly outperforms the Gaussian noise; LMO-DP performs even better for smaller $\epsilon$ since the avearge reduction rate slightly declines as $\epsilon$ increases.

**Ablation Study of LMO-DP Noise**. Since the LMO-DP noise is a Laplace-based two-fold noise (the first-fold is the Laplace distribution while the second-fold is a mixture distribution of three PDFs), we conducted an ablation study for it. Specifically, the "inverse" of "scale parameter" of the Laplace distribution (1/b) is subject to the linear combination of Gamma, Exponential and Uniform distributions. Our ablation study in Appendix F shows that the Uniform distribution (in the second-fold mixture distribution) contributes most to the sub-optimal performance of LMO-DP noise.

## 5 EXPERIMENTS

We experimentally evaluate our LMO-DP on fine-tuning with differential privacy. The experiments are performed on a spectrum of tasks including sentence classification (Wang et al. (2018)) and table-to-text generation (Yu et al. (2021)), utilizing RoBERTa-base, RoBERTa-large (Liu et al. (2019)), BERT-base, and BERT-large (Devlin et al. (2018)) and GPT-2 models (Yu et al. (2021)). The results highlight the versatility and universality of LMO-DP across different tasks and model structures.

### 5.1 EXPERIMENTAL SETTING

**Sentence Classification.** We first evaluate the LMO-DP on the sentence classification task from the GLUE benchmark. Specifically, following Yu et al. (2021), we fully fine-tune the RoBERTa-base, RoBERTa-large (Liu et al. (2019)), BERT-base and BERT-large (Devlin et al. (2018)) models on MNLI, SST-2, QNLI, and QQP datasets (Wang et al. (2018)). These datasets are widely used in evaluating private training. SST-2 has more than 60k+ samples in the training set; QNLI has more than 100k+ samples; MNLI and QQP contain more than 350k but less than 400k samples for each dataset. SST-2, QNLI, and QQP include two classes each; MNLI includes three classes.

**Table-to-Text Generation.** We also evaluate the LMO-DP on the table-to-text generation task that generates the descriptions of table entries. We fine-tune the GPT-2 model (Yu et al. (2021)) with the E2E dataset (Novikova et al. (2017)).

Most SOTA methods tend to achieve high accuracy in case of large total privacy budgets $\epsilon$ (weak privacy regimes). Specifically, when $\epsilon \geq 3$, they can deliver accuracy around $93\% \sim 94\%$. However, it remains unclear how these models perform when subjected to stronger DP guarantees, characterized by a total $\epsilon < 3$. In this work, we comprehensively evaluate the model performance for a wider range of $\epsilon$ (including $< 3$), and we compare LMO-DP with the SOTA methods (DP-SGD variants) on the accuracy and convergence with the same total privacy budget.

We conducted our experiments on two servers: Intel(R) Xeon(R) Platinum 8336C CPU @ 2.30GHz, 2T RAM, and 8×NVIDIA A100 SXM4 80G GPUs, and AMD Ryzen Threadripper PRO 5975WX 32-Cores CPUs, 500G RAM, 3×NVIDIA Quadro RTX A6000 48GB GPUs. Our souce code is available at `https://github.com/takko0234/lmo-dp` anonymously.

## 5.2 PERFORMANCE ON SENTENCE CLASSIFICATION

**Hyperparameter Setting**. LMO-DP ensures $\epsilon$-DP with $\delta = 0$, and thus we set $\delta = 10^{-10}$ for the baseline DP-SGD (Li et al. (2022)) for relatively fair comparisons. We first select different privacy parameters at each iteration of the fine-tuning: $\{0.3, 0.7, 2, 3\}$, and the total privacy loss $\epsilon$ will be derived with composition. The optimal weight values for three PDFs $a_1$, $a_2$, and $a_3$ are in the range $[0.1, 0.3]$ (varying on different $\epsilon$). Additionally, we set clipping threshold for gradients as 1, across all methods. We employ a batch size of 2048 and 6 training epochs. Consequently, the sampling rate for the training data is calculated as $\frac{2048}{|D|}$, where $|D|$ denotes the size of the dataset.

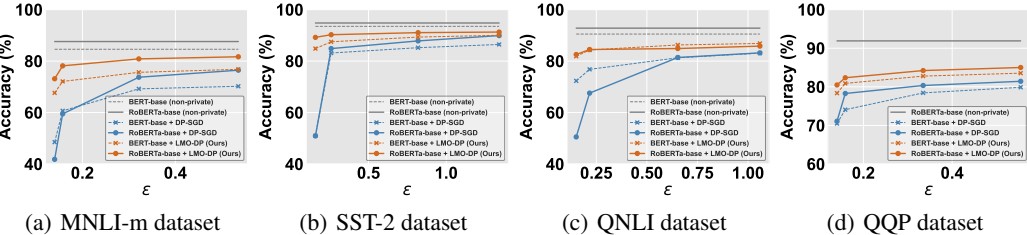

Figure 2: Accuracy of sentence classification task for BERT-base and RoBERTa-base models (100M parameters). For larger $\epsilon$, the results of SOTA are approximating LMO-DP, and they are not plotted.

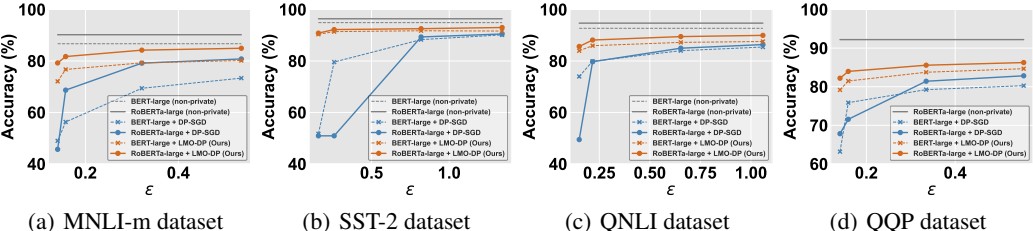

Figure 3: Accuracy of sentence classification task for BERT-large and RoBERTa-large (300M parameters). For larger $\epsilon$, the results of SOTA are approximating LMO-DP, and they are not plotted.

As shown in Figure 2 and 3, while the x-axis shows the total privacy loss $\epsilon$, and the y-axis shows the accuracy given $\epsilon$ (after composition) for fine-tuning. We compare the results of non-private fine-tuning, DP-SGD, and LMO-DP. Figure 2 demonstrates the results for the BERT-base and RoBERTa-base models (100M parameters) over four datasets. When $\epsilon$ increases, the accuracy also increases. Compared with DP-SGD, our results show better accuracy and have significant improvement under small $\epsilon$ (e.g., 0.15-0.5). Even with this small overall privacy loss, our LMO-DP can still have the prediction accuracy close to the non-private results, achieving over 80% accuracy. We observe similar trends in Figure 3 which reports the results for the BERT-large and RoBERTa-large models (300M parameters) over four datasets. LMO-DP still exhibits much better accuracy than DP-SGD.

To evaluate the convergence of different methods, the y-axis in Figure 4 means the steps taken when achieving the same accuracy (varying in each experiment) on varying privacy $\epsilon$.[4] Figure 4 shows the number of steps with four models over the MNLI-m and QQP datasets. Clearly, LMO-DP needs less steps to achieve the same reasonably good accuracy. The number of steps can be reduced by 50% on the MNLI-m dataset when $\epsilon$ is less than 0.3 and the number of steps of DP-SGD on the QQP dataset is always much higher than LMO-DP. This demonstrates that LMO-DP converges faster than DP-SGD. We show the training process for 6 epochs in Appendix E.2.

---

[4]We cannot compare the steps to achieve the highest/converged accuracy since the baseline(s) cannot achieve the same high accuracy as LMO-DP (given the same small privacy budget). Thus, we found the minimum of the highest accuracy for all the tasks on a specific dataset using similar-parameter models as the reference; then we count the steps to reach this reference. For instance, we evaluate the steps that need to reach 63.18% accuracy and 67.84% accuracy for the BERT-large and RoBERTa-large on QQP dataset.

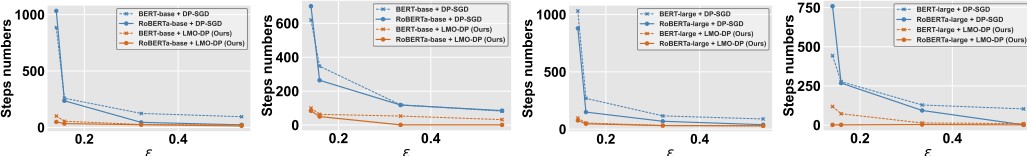

(a) MNLI-m dataset (base)   (b) QQP dataset (base)   (c) MNLI-m dataset (large)   (d) QQP dataset (large)

Figure 4: Steps to achieve the same accuracy (varying in each experiment). For larger $\epsilon$, the results of SOTA methods are approximating LMO-DP (converging quickly), and they are not plotted.

### 5.3 PERFORMANCE ON TABLE-TO-TEXT GENERATION TASK

**Hyperparameter Setting**. Next, we investigate the DP fine-tuning for text generation tasks using the GPT-2 model over the SST-2 dataset. We evaluate LMO-DP with $\delta = 0$ and DP-SGD (Li et al. (2022)) under a fixed $\delta = 8 \cdot 10^{-6}$ (same setting as Li et al. (2022)). We also apply the same settings of privacy budget at each iteration, weights and clipping threshold as the sentence classification. We only employ a batch size of 16 which causes the total privacy budget to be less than $0.2$ (after composition).

Table 2 presents the results with five different metrics by following Yu et al. (2021). We observe that LMO-DP yields results that are more closely aligned with the non-private re-

Table 2: Fine-tuning GPT-2 on the E2E dataset.

| Total $\epsilon$ | Methods | Metrics | | | | |
|---|---|---|---|---|---|---|
| | | BLEU | NIST | METEOR | ROUGE-L | CIDEr |
| 0.046807 | DP-SGD | 21.36 | 2.3185 | 0.3575 | 55.99 | 0.66 |
| | **LMO-DP** | **30.82** | **3.2484** | **0.3622** | **59.19** | **1.3826** |
| 0.046870 | DP-SGD | 25.47 | 2.5943 | 0.4103 | 60.28 | 0.8656 |
| | **LMO-DP** | **43.1** | **4.3517** | **0.441** | **69.7** | **2.0162** |
| 0.067623 | DP-SGD | 39.66 | 4.0486 | 0.433 | 67.56 | 1.836 |
| | **LMO-DP** | **49.91** | **5.3452** | **0.4495** | **68.94** | **3.073** |
| 0.176436 | DP-SGD | 44.32 | 4.471 | 0.4429 | 70.5 | 2.2172 |
| | **LMO-DP** | **53.51** | **5.7178** | **0.4489** | 68.87 | **3.3614** |
| Non-private | | 54.25 | 6.4832 | 0.4709 | 68.7 | 3.9951 |

sults (larger values of all these metrics exhibit more accurate generated texts). It is worth noting that the improvement can be up to $50\%$ on some metrics (e.g., CIDEr). The ROUGE-L of both LMO-DP and DP-SGD can be slightly higher than the original values (the last pair of results) since they are both fine-tuned based on the same LMs with rich vocabulary and downstream dataset and thus have a greater chance of generating approximate texts.

### 5.4 INTEGRATION WITH GHOST CLIPPING AND COMPARISON WITH SOTA METHODS

Recall that some existing methods focus on memory reduction (e.g., Ghost Clipping (Li et al. (2022))) or parameter efficiency (Yu et al. (2021)). Since LMO-DP is orthnogal to them, w.l.o.g., we also evaluate the accuracy for the integration with the Ghost Clipping (Li et al. (2022)) and compare it with SOTA methods (similarly, using the same RoBERTa-large model on the SST-2 dataset). As $\epsilon > 3$, the best accuracy of SOTA methods and LMO-DP would be close to each other, thus $\epsilon$ is set to not exceed 3 ($\delta$ is set to be close to 0, e.g., $10^{-10}$ while $\delta = 0$ for LMO-DP). Table 3

Table 3: Accuracy (%) for LMO-DP (Ghost Clipping Mode) vs. SOTA (RoBERTa-large on SST-2 for sentiment classification).

| Method | Total $\epsilon$ | | | | | |
|---|---|---|---|---|---|---|
| | 0.16 | 0.3 | 0.9 | 1.4 | 3 | $\infty$ |
| Li et al. (2022) | 50.92 | 50.92 | 89.33 | 90.48 | 91.06 | 96.20 |
| Yu et al. (2021) | - | - | 51.31 | 51.31 | 51.31 | 96.40 |
| Bu et al. (2022b) | 49.08 | 49.43 | 50.92 | 50.92 | 54.58 | 95.50 |
| He et al. (2022) | - | - | - | - | 93.87* | 96.20* |
| Bu et al. (2023) | 49.08 | 50.92 | 87.72 | 90.02 | 90.14 | - |
| **LMO-DP** | **90.83** | **92.20** | **92.55** | **93.00** | **93.92** | **96.20** |

shows the results of LMO-DP (Ghost Clipping mode) compared to five SOTA methods.[5] LMO-DP exhibits superior accuracy over all the baselines, especially for very strong DP settings.

### 6 CONCLUSION

In this paper, we propose a Language Model-based Optimal Differential Privacy (LMO-DP) framework, allowing for accurately fine-tuning language models even in very strict privacy settings. LMO-DP outperforms existing methods (DP-SGD and its variants) on both accuracy and convergence, as demonstrated in the experiments with multiple language models on multiple datasets.

---

[5]He et al. (2022) is not open-sourced. We include its result (*) for $\epsilon = 3$, $\delta = 1/n^{1.1}$ given training size $n$.

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

# A DEFINITIONS

## A.1 PROBABILITY SPACE DEFINITION

Let us consider a fixed probability space $(\Omega, \mathcal{F}, \mathbb{P})$. Let $\mathcal{D}$ be a space of dataset. A mechanism is defined as a mapping $\mathcal{M} : \mathcal{D} \times \Omega \to \mathcal{R}$, where $\mathcal{R}$ is a measurable output space with $\sigma$-algebra $\mathcal{M}$. For any element $D \in \mathcal{D}$, $\mathcal{M}(D, \cdot)$ is a random variable, typically denoted as $\mathcal{M}(D)$. A mechanism can be interpreted as a probabilistic algorithm used to provide responses to a query $q$, which is itself a mapping $q : \mathcal{D} \to \mathcal{R}$. In some cases, we explicitly index the mechanism by the specific query of interest, denoting it as $\mathcal{M}_q$.

## A.2 MOMENT GENERATING FUNCTION AND LINEAR COMBINATION OF MGFS

**Definition 2** *The moment-generating function of a random variable $x$ is $M_X(t) := \mathbb{E}\left[e^{tX}\right], t \in \mathbb{R}$ wherever this expectation exists. The moment-generating function is the expectation of the random variable $e^{tX}$ (Walker (1965)).*

**Theorem A.1 (MGF of Linear Combination of RVs)** *If $x_1, \cdots, x_n$ are $n$ independent random variables (RVs) with MGFs $M_{x_i}(t) = \mathbb{E}(e^{tx_i})$ for $i = 1, \cdots, n$, then the MGF of the linear combination $Y = \sum_{i=1}^{n} a_i x_i$ is $\prod_{i=1}^{n} M_{x_i}(a_i t)$.*

## A.3 R²DP MECHANISM

**Theorem A.2** *For a given query function $q : \mathcal{D} \to \mathbb{R}$, any measurable subset $S \subset \mathbb{R}$, and a dataset $D \in \mathcal{D}$, consider a randomized mechanism $\mathcal{M}_q(D, b) : \mathcal{D} \times \Omega \to \mathbb{R}$ defined as $\mathcal{M}_q(d) = q(d) + w$, where $w \sim Lap(b)$ and $1/b \sim f \in \mathcal{F}$. Then, the probability that $\mathcal{M}_q(D, b)$ falls within subset $S$ can be expressed as:*

$$\mathbb{P}(\mathcal{M}_q(D, b) \in S) = \frac{1}{2} \cdot \left[ -M_{1/b}(-|x - q(D)|) \cdot \mathbb{1}_{\{S \geq q(D)\}} + M_{1/b}(-|x - q(D)|) \cdot \mathbb{1}_{\{S < q(D)\}} \right] \quad (5)$$

*where $M_f(t)$ denotes the moment-generating function (MGF) of the random variable $f$, and $\mathbb{1}$ is the indicator function (Proof in Mohammady et al. (2020), Appendix C).*

# B PROOFS

## B.1 PROOF OF THEOREM 4.1

For any $\alpha > 1$ and for any $\tilde{\mathbb{P}}^D_{i/o}$ in the set $\mathcal{F}$, we want to minimize the difference between the Rényi-DP of $\tilde{\mathbb{P}}^D_{i/o}$, denoted as $\epsilon^{\tilde{\mathbb{P}}^D_{i/o}}$, and a predefined value $\epsilon_\alpha$ representing Rényi-DP of the optimal geometry $X^D$. To achieve this, we formulate a series of optimization problems. These problems are parametrized by $\alpha$ and seek to minimize the following quantity:

$$\left\| \frac{1}{\alpha - 1} \log \frac{1}{N} \left( \sum_{i=1}^{N} \sum_{o=1}^{C} \left( \tilde{\mathbb{P}}^D_{i/o} \right)^\alpha \cdot \left( \tilde{\mathbb{P}}^{D'}_{i/o} \right)^{1-\alpha} \right) + \epsilon_\alpha \right\|.$$

The optimization is performed for all $\alpha > 1$ and for fixed, given $\epsilon_\alpha$. Now, let's apply Abel's inequality to the quantity in the optimization problem. Abel's inequality provides a bound on the absolute value of the inner product of two sequences in special cases. For non-increasing and non-negative sequences $\{a_n\}$ and $\{b_1, b_2, \cdots\}$ of real or complex numbers:

$$(\min_r \sum_{i=1}^{r} b_i) a_1 \leq \sum_i a_i b_i \leq (\max_r \sum_{i=1}^{r} b_i) a_1$$

We can apply this inequality to the sum inside the logarithm in our optimization problem. WLOG suppose for all $i$, $\tilde{\mathbb{P}}^D_{i/1} \geq \tilde{\mathbb{P}}^D_{i/2} \geq \cdots \geq \tilde{\mathbb{P}}^D_{i/C}$, then:

$$\sum_{i=1}^{N} (\tilde{\mathbb{P}}^D_{i/1})^\alpha \left( \min_r \sum_{o=1}^{r} \left( \tilde{\mathbb{P}}^{D'}_{i/o} \right)^{1-\alpha} \right) \leq \sum_{i=1}^{N} \sum_{o=1}^{C} \left( \tilde{\mathbb{P}}^D_{i/o} \right)^\alpha \cdot \left( \tilde{\mathbb{P}}^{D'}_{i/o} \right)^{1-\alpha} \leq \sum_{i=1}^{N} (\tilde{\mathbb{P}}^D_{i/1})^\alpha \left( \max_r \sum_{o=1}^{r} \left( \tilde{\mathbb{P}}^{D'}_{i/o} \right)^{1-\alpha} \right)$$

Usually, the highest probability, denoted as $\tilde{\mathbb{P}}^D_{i/1}$, represents the predicted class probability, and in this context, it's equal to $\mathbb{P}^D_{i/o^c}$. We use minimum and maximum functions in the lower and upper bound calculations, respectively. This choice is based on the fact that $\tilde{\mathbb{P}}^{D'}_{i/o}$ always falls within the range $[0, 1]$. Specifically, the minimum function is set to 1, and the maximum function approaches infinity. Because of these characteristics, we ignore the upper bound, and we can simplify the lower bound to $\sum_{i=1}^{N} (\tilde{\mathbb{P}}^D_{i/o^c})^\alpha$. Thus

$$\epsilon_\alpha + \log \left( \frac{1}{N} \sum_{i=1}^{N} \sum_{o=1}^{C} \left( \tilde{\mathbb{P}}^D_{i/o} \right)^\alpha \cdot \left( \tilde{\mathbb{P}}^{D'}_{i/o} \right)^{1-\alpha} \right) \leq \epsilon_\alpha + \log \left( \frac{1}{N} \sum_{i=1}^{N} \tilde{\mathbb{P}}^D_{i/o^c} \right),$$

The left-hand side, which represents the expression in our optimization problem, should ideally approach zero. This occurs when $\tilde{\mathbb{P}}^D_{i/o}$ converges to $X^D$. In a similar fashion, in this scenario, the upper bound will be accurately defined as the optimal cross-entropy term:

$$-\log \left( \frac{1}{N} \sum_{i=1}^{N} X^D_{i/o^c} \right).$$

This demonstrates that $\tilde{\mathbb{P}}^D_{i/o}$ is a sub-optimal solution for problem (2), as it forms a lower bound for the quantity in the optimization problem.

## B.2  PROOF OF THEOREM 4.2

In our framework, the mixture distributions include the Gamma distribution $Y_1 \sim \Gamma(k, \theta)$, Exponential distribution $Y_2 \sim Exp(\lambda)$, and Uniform distribution $Y_3 \sim U(a, b)$. The $\epsilon$-Rényi differential privacy for order $\alpha > 1$ can be computed as follows:

$$\epsilon_{\text{LMO-DP}} = \frac{1}{\alpha - 1} \log \left[ \frac{\alpha \prod_{i=1}^{3} M_{f_i}(a_i(\alpha - 1)) + (\alpha - 1) \prod_{i=1}^{3} M_{f_i}(a_i(-\alpha))}{2\alpha - 1} \right]$$

where $a_1(\alpha - 1) < \frac{1}{\theta}$, $a_2(\alpha - 1) < \lambda$, $b > a$, $k > 0$, $\theta > 0$, and $\lambda > 0$.

Here, $M_{f_i}$ denotes the Moment Generating Function of the mixture components, and $a_1$, $a_2$, and $a_3$ are the weights for the Gamma, Exponential, and Uniform distributions, respectively. The differential privacy is converted from Rényi-DP to $(\epsilon, \delta)$-DP.

## C  QUANTIFYING THE LMO SEARCH SPACE: A PROPOSED APPROACH

In our quest to understand the LMO space, we introduce the "Comprehensiveness Explorer." This algorithm probes the LMO space's diversity and capability to represent various probability density functions.

### C.1  THE KEY TOOL: SIMULATED SEARCH SPACE

At the core of the Comprehensiveness Explorer is the simulated search space, constructed using the Multinomial Probability Density Function (PDF). It enables us to quantitatively assess the richness of the LMO space. The Multinomial PDF, denoted as Multinomial($N, k, p = 1/k$), models the scenario of distributing a unit probability mass uniformly among $k$ classes, with precision controlled by parameter $q$. We sample from this space to capture probability distributions under varying quantization rates $q$ and domain sizes $k$.

### C.2  QUANTIFYING COMPREHENSIVENESS

Algorithm 2 (Quantification of LMO Search Space) is employed to assess the comprehensiveness of this search space in comparison to a universally simulated space, introducing a novel quantification test. We utilize three distance metrics (both probabilistic and deterministic): KL divergence, $\ell_2$ and EMD metrics to measure the distance between these two spaces. Our results demonstrate that, for any given noise 'n', there exists an LMO noise that remains close to 'n'. Please note for the EMD metric, we experience a small divergence (scaled by $10^{-3}$) when the domain of noise increases which necessitates a smaller quantization rate to achieve even better results.

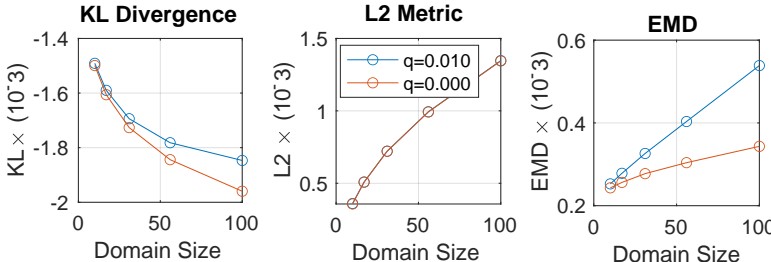

Figure 5: Generated noise using the LMO space exhibits a remarkable level of comprehensiveness concerning three distance metrics. These results are derived from the quantification of the LMO space, as outlined in Algorithm 2.

### C.3 ADAPTIVE LMO SAMPLE GENERATION

To ensure LMO samples resemble the simulated space, we adjust LMO distribution parameters based on simulated space statistics. Specifically, results generated in Figure 5 are for $q$ and $k$ fine-tuned using the following steps: 1) estimate $\mu_{\text{sim}}$ and $\sigma_{\text{sim}}^2$, 2) adjust $\mu_{\text{LMO}}$ to match $\mu_{\text{sim}}$, 3) adjust $\sigma_{\text{LMO}}$ based on $\sigma_{\text{sim}}^2$ and 4) generate LMO samples with adjusted parameters.

This adaptive approach ensures LMO samples match the simulated space's statistical properties.

## D ALGORITHMS

---

**Algorithm 2:** Quantification of LMO Search Space

---

**Input:** $\mathcal{Q}$: quantization values; $\mathcal{K}$: domain size; $M$: sampling times; $\mathcal{S}$: searched ranges of distribution parameters and their weights; $\ell$: distance metric (e.g., KL-divergence, $\ell_2$ distance)
    // $\mathcal{Q} = \left[10^{-1}, 10^{-2}, \cdots, 10^{-q}\right]$; $q, \mathcal{K}, N, M$ are typically large numbers.
**Output:** Distance $\mathcal{D}$ between the universal search space and LMO space.
    /* Simulating universal search space                                    */
1 **foreach** $q$ in $\mathcal{Q}$ **do**
2    **foreach** $k$ in $\mathcal{K}$ **do**
3       **foreach** $i \in [1, M]$ **do**
4          $N = 1/q, p = 1/k$
5          $x_i \sim \text{Multinomial}(N, k, p)$
          /* Generating LMO search space                              */
6       **foreach** $j \in [1, M]$ **do**
7          $y_j \sim \text{Lap}(0, b(\mathcal{S}), k)$
8       $\mathcal{D}_{q,k} = \ell(x, y)/M$
9 **return** $\mathcal{D}$

---

**Algorithm 3:** Rényi-DP Accountant Optimization $F_1$

---

**Input:** $A_1$: privacy budget $\{\epsilon, \delta\}$, $A_3$      // $A_3$={(Gamma, Exponential, Uniform), $\alpha_{max}$, $\mathcal{S}$}
**Output:** LMO parameters $\mathcal{S}_{\text{opt}}$          // Best LMO mechanism
    /* Step 1:  Defining the MGF of distributions                          */
1 **if** *Gamma* $\in A_3$ **then**
2    $M_{Y_1}(t) \leftarrow (1 - t\theta)^{-k}$, $t < \frac{1}{\theta}$
3 **else**
4    $M_{Y_1}(t) \leftarrow 0$
5 **if** *Exponential* $\in A_3$ **then**
6    $M_{Y_2}(t) \leftarrow \left(1 - t\lambda^{-1}\right)^{-1}$, $t < \lambda$
7 **else**
8    $M_{Y_2}2(t) \leftarrow 0$
9 **if** *Uniform* $\in A_3$ **then**
10    $M_{Y_3}(t) \leftarrow \dfrac{e^{tb} - e^{ta}}{t(b - a)}$
11 **else**
12    $M_{Y_3}(t) \leftarrow 0$
13 $M_Y(t) \leftarrow a_1 \cdot M_{Y_1}(t) + a_2 \cdot M_{Y_1}(t) + a_3 \cdot M_{Y_3}(t)$
    /* Step 2:  Finding the optimal $S_{opt}$ by grid search                */
14 **foreach** $S \in \mathcal{S}$ **do**
15    $\alpha_1, \alpha_2, \alpha_3, k, \theta, \lambda, b, a = S$
       /* Rényi-DP Accountant Optimization (Theorem F.4 in Mohammady
         et al. (2020))                                              */
16    **foreach** $\alpha \in [2, \alpha_{max}]$ **do**
17       $\epsilon_{\text{Rényi},\alpha} = \frac{1}{\alpha-1} \log \left[\frac{\alpha M_Y(\alpha-1) + (\alpha-1)M_Y(-\alpha)}{2\alpha-1}\right]$
18       $\epsilon'$.append(($\mathcal{G}(\alpha, \epsilon_{\text{Rényi},\alpha}, \delta)$))      // $\mathcal{G}$: convert Rényi-DP to DP
19    **if** *max($\epsilon'$)* $< \epsilon$ **then**
20       $S_{\text{opt}} \leftarrow S$
21 **return** $S_{\text{opt}}$

---

# E ON QUALITY OF LMO-DP NOISE

## E.1 ADDITIONAL RESULTS FOR LMO-DP NOISE VS GAUSSIAN NOISE

Figure 6(a) plots the simple entropy and Figure 6(b) plots the variance of the Gaussian and LMO-DP noises. First, we generated the Gaussian noise and LMO-DP noises which have the exact privacy cost for a single noise (same $\epsilon$ value). Then, we computed the histogram and probability density function of these sampled noises. Finally, we plot the simple entropy and variance of the Gaussian and LMO-DP noises for specific $\epsilon$ values. These comparisons exhibit that our LMO-DP noises have lower entropy and variance when compared to Gaussian noises with the same privacy budget. Moreover, in Appendix C, we propose a novel approach to quanitify the extent of the LMO search space via first simulating the entire search space with different quantization rates and domain sizes and then measure the worst-case distance of LMO search space to the simulated space, using both probabilistic "KL-divergence" and deterministic "$\ell_2$" distance metrics.

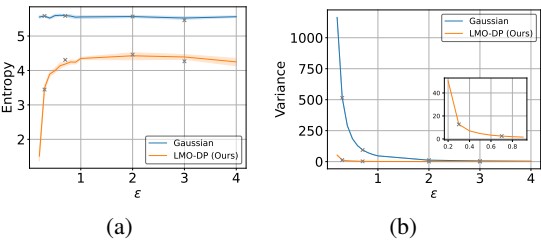

Figure 6: (a) The simple entropy comparison of Gaussian and LMO-DP noises. (b) The variance comparison of Gaussian and LMO-DP noises.

## E.2 DETAILS OF TRAINING PROCESS

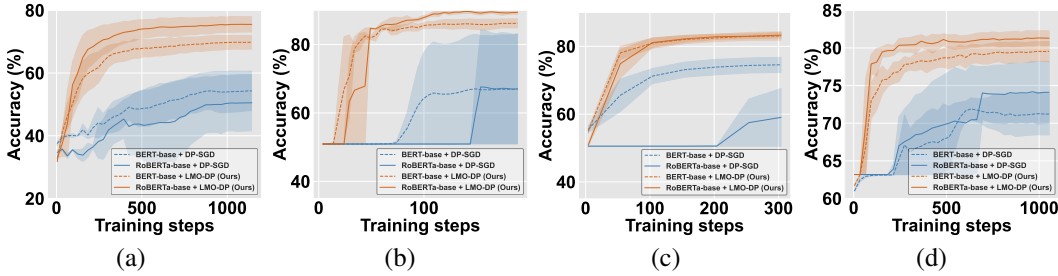

Figure 7: The training process of sentence classification task for BERT-base and RoBERTa-base (using 100M parameters) with small privacy budget ($\epsilon$=0.2 or $\epsilon$=0.3). (a) MNLI-m dataset; (b) SST-2 dataset; (c) QNLI dataset; (d) QQP dataset.

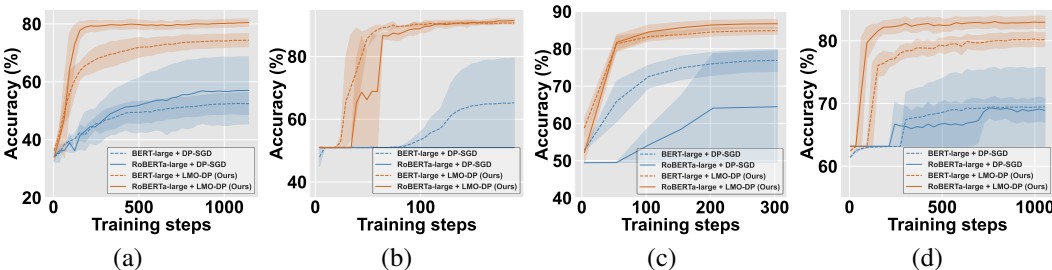

Figure 8: The training process of the sentence classification task for BERT-large and RoBERTa-large (using 300M parameters) with small privacy budget ($\epsilon$=0.2 or $\epsilon$=0.3). (a) MNLI-m dataset; (b) SST-2 dataset; (c) QNLI dataset; (d) QQP dataset.

# F  ABLATION STUDY

## F.1  ONE DISTRIBUTION VS. MIXTURE DISTRIBUTION IN LMO-DP

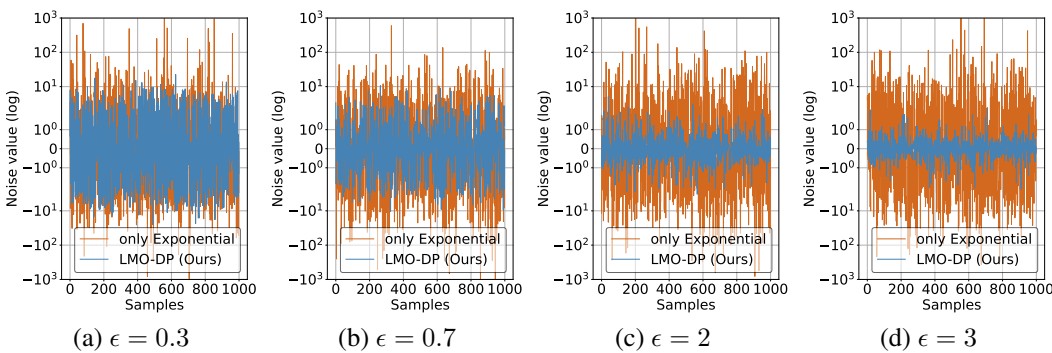

(a) $\epsilon = 0.3$     (b) $\epsilon = 0.7$     (c) $\epsilon = 2$     (d) $\epsilon = 3$

Figure 9: Exponential distribution vs mixture distribution (with the same remaining setting). The noise generated by the mixture distribution (as the second-fold) in LMO-DP is significantly smaller than that replaces the mixture distribution with the Exponential distribution, especially $\epsilon = 2$ or 3.

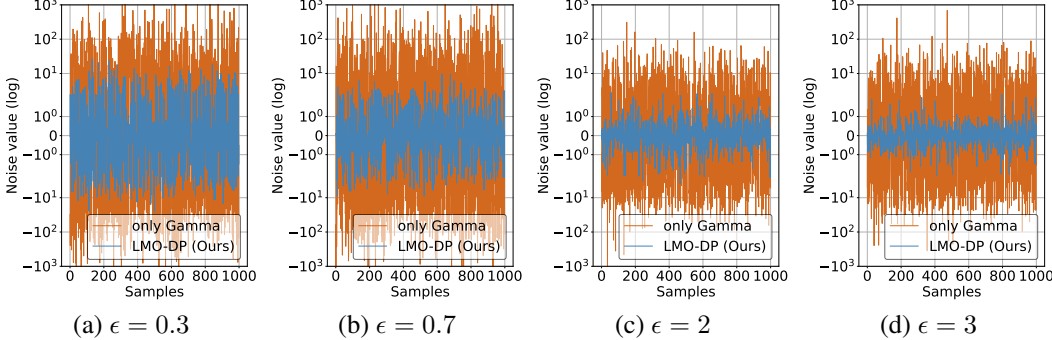

(a) $\epsilon = 0.3$     (b) $\epsilon = 0.7$     (c) $\epsilon = 2$     (d) $\epsilon = 3$

Figure 10: Gamma distribution vs mixture distribution (with the same remaining setting). The noise generated by the mixture distribution (as the second-fold) in LMO-DP is significantly smaller than that replaces the mixture distribution with the Gamma distribution for all $\epsilon$.

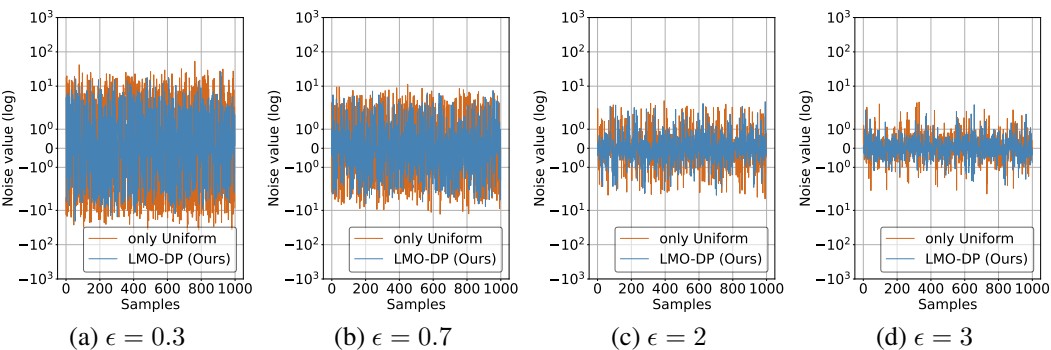

(a) $\epsilon = 0.3$     (b) $\epsilon = 0.7$     (c) $\epsilon = 2$     (d) $\epsilon = 3$

Figure 11: Uniform distribution vs mixture distribution (with the same remaining setting). The noise generated by the mixture distribution (as the second-fold) in LMO-DP is slightly smaller than that replaces the mixture distribution with the Uniform distribution. **The results demonstrate that Uniform distribution contributes more to the subopitmal noise**.

## F.2  MIXTURE OF TWO DISTRIBUTIONS VS. MIXTURE OF THREE DISTRIBUTION IN LMO-DP

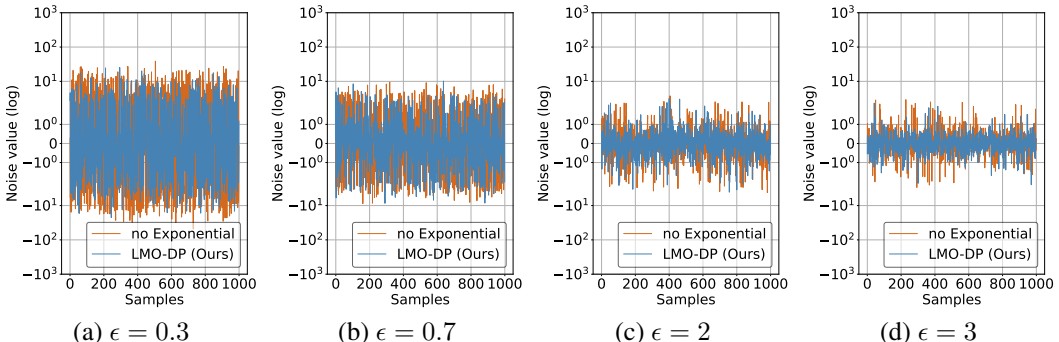

|                |                |              |              |
|:--------------:|:--------------:|:------------:|:------------:|
| (a) $\epsilon = 0.3$ | (b) $\epsilon = 0.7$ | (c) $\epsilon = 2$ | (d) $\epsilon = 3$ |

Figure 12: Mixture of Gamma and Uniform distributions vs mixture of three distribution (with the same remaining setting). The noise generated by the mixture of three distributions (as the second-fold) in LMO-DP is slightly smaller than that removes the Exponential distribution.

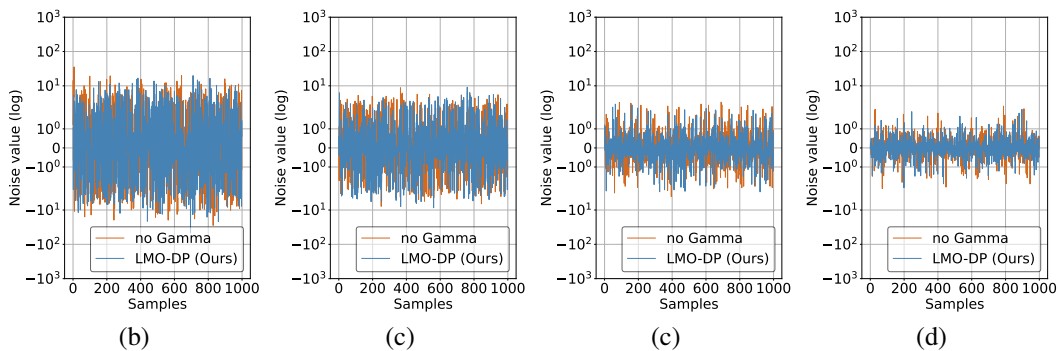

|      |      |      |      |
|:----:|:----:|:----:|:----:|
| (b)  | (c)  | (c)  | (d)  |

Figure 13: Mixture of Exponential and Uniform distributions vs mixture of three distribution (with the same remaining setting). The noise generated by the mixture of three distributions (as the second-fold) in LMO-DP is slightly smaller than that removes the Gamma distribution.

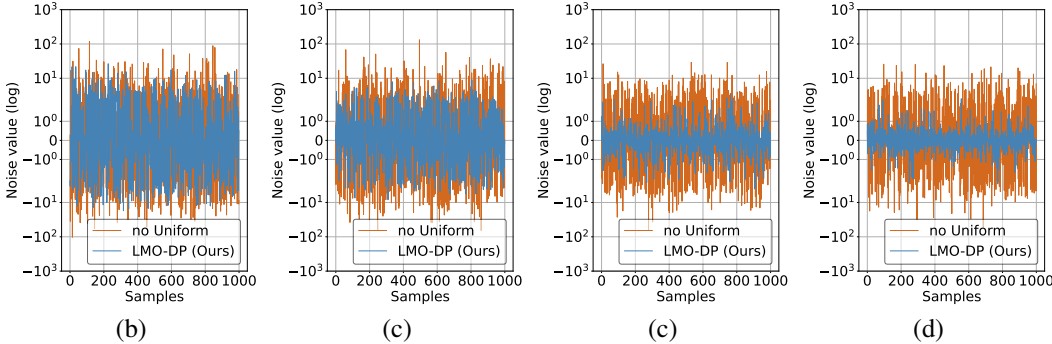

|      |      |      |      |
|:----:|:----:|:----:|:----:|
| (b)  | (c)  | (c)  | (d)  |

Figure 14: Mixture of Gamma and Exponential distributions vs mixture of three distribution (with the same remaining setting). The noise generated by the mixture of three distributions (as the second-fold) in LMO-DP is smaller than that removes the Uniform distribution, especially for large $\epsilon$. **The results again demonstrate that Uniform distribution contributes more to the subopitmal noise**.

