# A SUPPLEMENTARY DOCUMENT

## A.1 LMO-DP NOISE VS GAUSSIAN NOISE

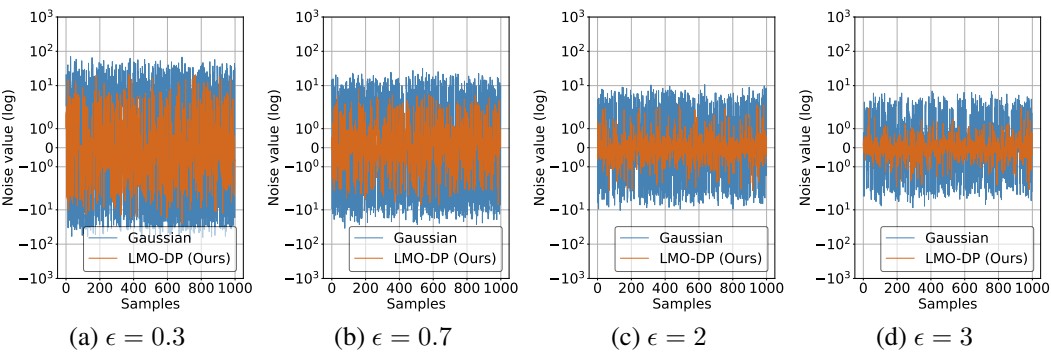

(a) $\epsilon = 0.3$     (b) $\epsilon = 0.7$     (c) $\epsilon = 2$     (d) $\epsilon = 3$

Figure A.1: LMO-DP vs Gaussian. (a) average reduction rate $95.13\%$. (b) average reduction rate $92.19\%$. (c) average reduction rate $87.71\%$. (d) average reduction rate $87.31\%$. The results demonstrate that the LMO-DP noise significantly outperforms the Gaussian noise; LMO-DP performs even better for smaller $\epsilon$ since the avearge reduction rate slightly declines as $\epsilon$ increases.

## A.2 ABLATION STUDY

### A.2.1 ONE DISTRIBUTION VS. MIXTURE DISTRIBUTION IN LMO-DP

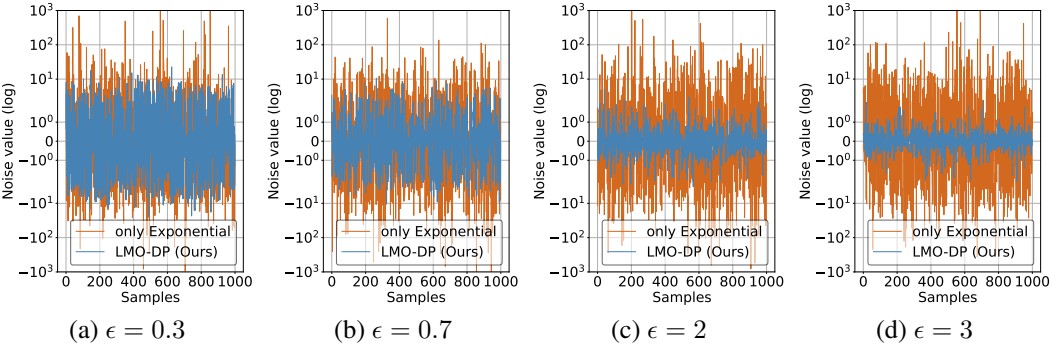

(a) $\epsilon = 0.3$     (b) $\epsilon = 0.7$     (c) $\epsilon = 2$     (d) $\epsilon = 3$

Figure A.2: Exponential distribution vs mixture distribution (with the same remaining setting). The noise generated by the mixture distribution (as the second-fold) in LMO-DP is significantly smaller than that replaces the mixture distribution with the Exponential distribution, especially $\epsilon = 2$ or 3.

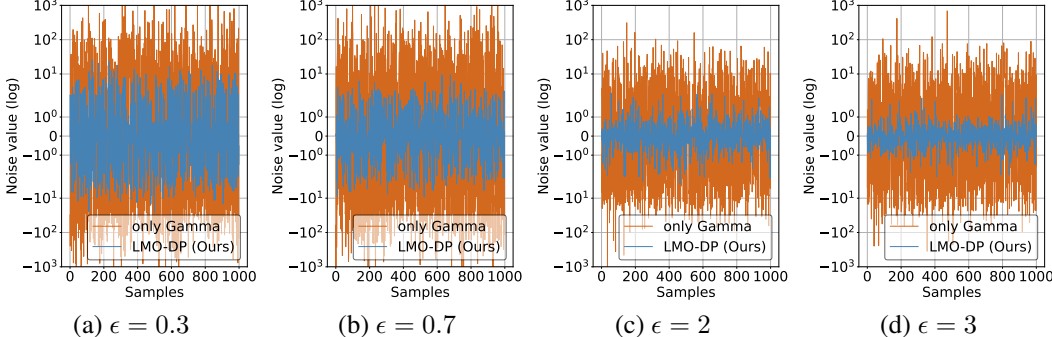

(a) $\epsilon = 0.3$     (b) $\epsilon = 0.7$     (c) $\epsilon = 2$     (d) $\epsilon = 3$

Figure A.3: Gamma distribution vs mixture distribution (with the same remaining setting). The noise generated by the mixture distribution (as the second-fold) in LMO-DP is significantly smaller than that replaces the mixture distribution with the Gamma distribution for all $\epsilon$.

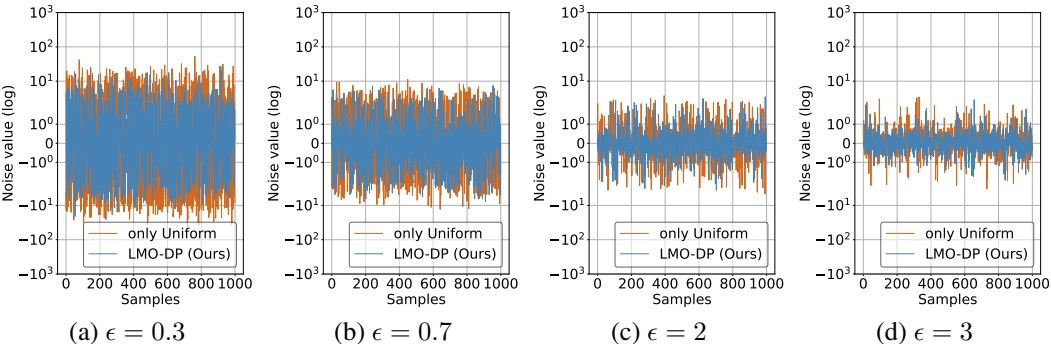

Figure A.4: Uniform distribution vs mixture distribution (with the same remaining setting). The noise generated by the mixture distribution (as the second-fold) in LMO-DP is slightly smaller than that replaces the mixture distribution with the uniform distribution. **The results demonstrate that uniform distribution contributes more to the subopitmal noise**.

### A.2.2 MIXTURE OF TWO DISTRIBUTIONS VS. MIXTURE OF THREE DISTRIBUTION IN LMO-DP

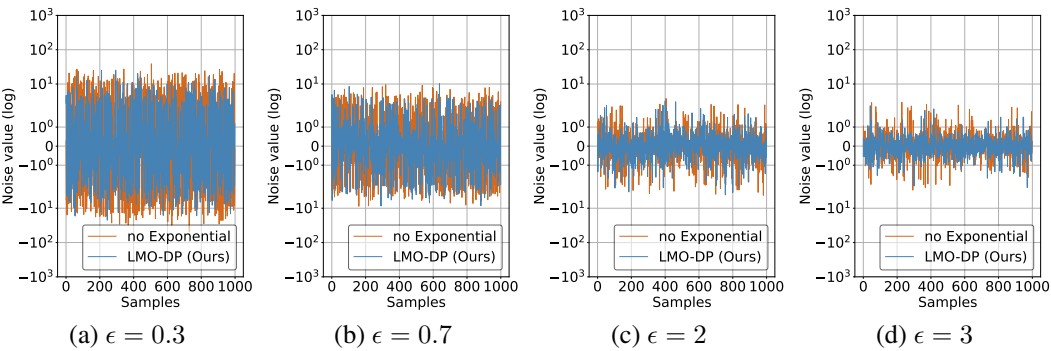

Figure A.5: Mixture of Gamma and uniform distributions vs mixture of three distribution (with the same remaining setting). The noise generated by the mixture of three distributions (as the second-fold) in LMO-DP is slightly smaller than that removes the exponential distribution.

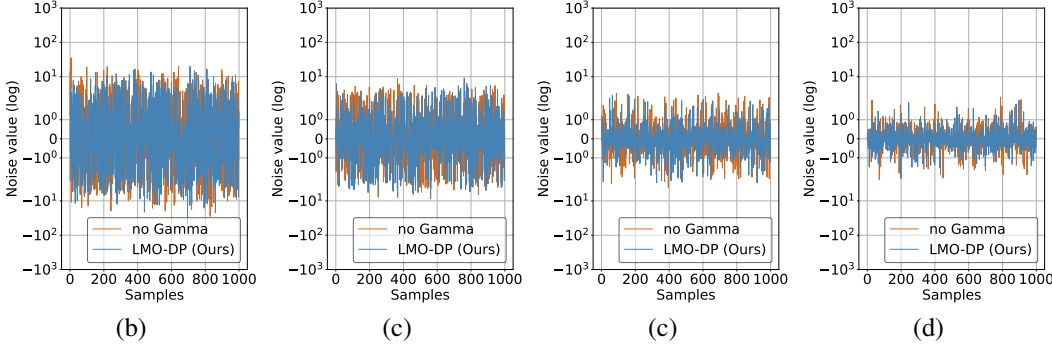

Figure A.6: Mixture of Exponential and uniform distributions vs mixture of three distribution (with the same remaining setting). The noise generated by the mixture of three distributions (as the second-fold) in LMO-DP is slightly smaller than that removes the Gamma distribution.

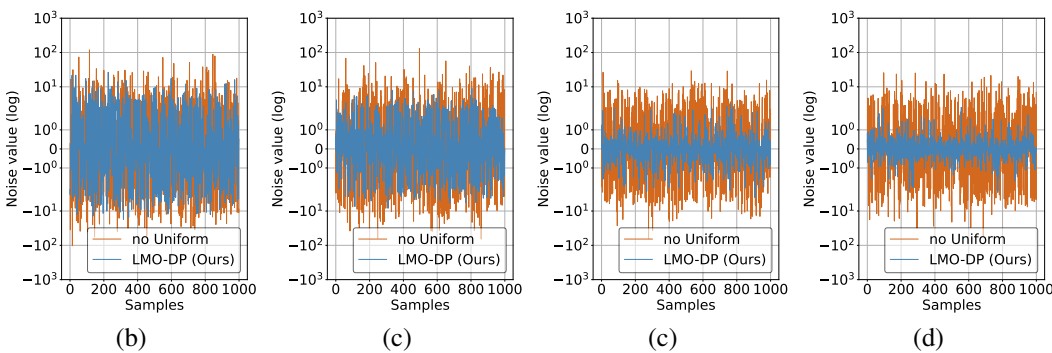

Figure A.7: Mixture of Gamma and exponential distributions vs mixture of three distribution (with the same remaining setting). The noise generated by the mixture of three distributions (as the second-fold) in LMO-DP is smaller than that removes the uniform distribution, especially for large $\epsilon$. **The results again demonstrate that uniform distribution contributes more to the subopitmal noise**.