# OpenReview forum: "LMO-DP: Accurately Fine-Tuning Language Models with Stronger Differential Privacy"
_ICLR.cc/2024/Conference — Submitted to ICLR 2024_

### Official Review · Reviewer_4F9v · 2023-10-31

**Soundness:** 2 fair
**Presentation:** 3 good
**Contribution:** 2 fair
**Rating:** 5
**Confidence:** 3

**Summary:**

This work proposes a new approach for fine-tuning large language models under differential privacy. It proposes an optimization framework that finds a good model parameter distribution subject to Renyi-DP constraints. Experiments show that the accuracy significantly surpasses existing works even with a very small privacy budget ($\varepsilon < 3$).

**Strengths:**

1. The problem of fine-tuning large models on private data is extremely relevant in practice.
2. The accuracy significantly surpasses previous work, especially in the high privacy regime.

**Weaknesses:**

1. There are many recent works on tight privacy accounting for differential privacy such as [1] and [2]. The authors should discuss whether they can be applied to the proposed algorithm.
2. In section 3.3, the global optimization seems to be formulated for a fixed pair of neighboring datasets $D$ and $D'$. However, differential privacy requires all pairs of $D$ and $D'$, so to me, there should be a max over all possible $D, D'$ somewhere in the formulation. I hope the authors can explain where the worst case over all neighboring datasets is considered in this formulation.


Comment on writing:
1. Many in-line citations do not have parenthesis and break the flow of the paper. For example: Section 2, DP-SGD Abadi et al. (2016). The reference should have a parenthesis around it.
2. Section 3.1: "We first define the differential privacy". "the" should be removed.

### References
[1] Zhu, Y., Dong, J. &amp; Wang, Y.. (2022).  Optimal Accounting of Differential Privacy via Characteristic Function . Proceedings of The 25th International Conference on Artificial Intelligence and Statistics<, in Proceedings of Machine Learning Research 151:4782-4817 Available from https://proceedings.mlr.press/v151/zhu22c.html.
[2] Individual Privacy Accounting with Gaussian Differential Privacy, Antti Koskela, Marlon Tobaben, Antti Honkela, ICLR 2023

**Questions:**

1. In equation (2), is there a particular reason for the range of $\alpha$ to be 2 to 129? Is it possible that the best parameter is achieved outside of this range for some datasets and problems?
2. It is surprising to me that the noise level for $\varepsilon =2,3$ is slightly larger than $\varepsilon=0.2$. Could the authors explain this phenomenon?

---

> ### Author Response · Authors · 2023-11-17
> **Rebuttal by Authors**
>
> Thanks a lot for the insightful and constructive comments. Please find our response to each comment as below.
>
> > Recent works on tight privacy accounting for differential privacy.
>
> Thanks for the suggestions (we will add a discussion and cite them). Their theories can also be applied to our proposed algorithm. We designed our LMO-DP method based on randomization which tries to find a smaller noise while meeting the same level of privacy guarantee. The contributions are orthogonal to these two works. If we integrate LMO-DP with these two works, we need to revise the constraints in Equation (2) and we can replace the RDP by those new accountants/methods with tighter bounds. New efficient and accurate solvers for finding the (sub)optimal noise would be desirable as well.
>
> > In section 3.3, the global optimization seems to be formulated for a fixed pair of neighboring datasets D and D′.
>
> Thanks a lot for this good catch. Yes, our design involves all pairs of D and D’ (worst case protection). It was a typo in the constraints in Equation (2). We will add a supremum over choice of D and D’ in Equation (2) to represent all the neighboring datasets (and worst case). Our proof (Theorem 4.1 in Appendix B.1) and implementation (in Algorithm 3 - LMO-DP in the original submission and the code) are indeed over the worst case, i.e., $\sup_{\forall D,D'\in\mathcal{D}}$.
>
> > Writing
>
> Thanks. We will fix them.
>
> > In equation (2), is there a particular reason for the range of \alpha to be 2 to 129? Is it possible that the best parameter is achieved outside of this range for some datasets and problems?
>
> The range of $\alpha$ was selected from 2 to 32 in most existing works/codes. However, when computing the R\'enyi privacy of the LMO mechanism within this $\alpha$ range, null values frequently arise. Expanding the upper bound of $\alpha$ can facilitate the computation of R\'enyi privacy for the LMO-DP mechanism. Nevertheless, opting for larger $\alpha$ values may result in moments too tiny for effective processing of the floating point numbers. Consequently, there exists a tradeoff between the choice of $\alpha$ and potential privacy leakage. For our specific tasks, we have empirically settled on 128 or 129 as a compromise. For other tasks, the selection of the $\alpha$ range should be based on privacy requirements and practical constraints (e.g., processing the floating point numbers).
>
> > It is surprising to me that the noise level for \epsilon=2,3 is slightly larger than \epsilon=0.2. Could the authors explain this phenomenon?
>
> Thanks for this question. The noise levels in our Figure 2 are different for $\epsilon=0.2$ and $\epsilon=2$. They look close to each other since different scales for y axes were applied in the subfigures. To show the dynamic of the LMO-DP mechanism, we replace Figure 2 with the statistical figure of sampled noise in the revised version (please see the supplementary document and the submission will be updated asap).

---

> > ### Comment · Reviewer_4F9v · 2023-12-04
> > **Scores unchanged.**
> >
> > Thank you for your response. I have read your response and the plots for the noise levels indeed look much nicer after the edit.
> >
> > However, as I read the proof of Theorem 4.1 and discussions with other reviewers, the private query to be protected seems to be the output labels/probabilities instead of gradients or model parameters. While this may also be useful in practice, (e.g. a chatbot answering questions from users), it is inconsistent with prior works (e.g. the baseline of Bu et.al. ICML 2023) based on DP-SGD which assume the adversary has access to intermediate gradient updates or final model output. This makes the comparison unfair. The authors should compare with similar works that only protect the final query, or modify their algorithms to protect gradient releases.

---

### Official Review · Reviewer_9q2i · 2023-11-01

**Soundness:** 3 good
**Presentation:** 3 good
**Contribution:** 3 good
**Rating:** 6
**Confidence:** 3

**Summary:**

The paper proposes a novel DP framework for training large language models. It employs the composition of sub optimal DP mechanisms for fine-tuning LLMs. The method empirically outperforms the existing baselines.

**Strengths:**

1. The paper provides a method to tightly compose the DP mechanisms compared to traditional gaussian mechanisms, which usually add a lot of noise.
2. LMO subspace is an interesting way of finding LM geometries, as it provides universal RDP guarantees along with strong empirical results.
3. Empirical results are strong compared to the baselines.
4. Algorithm is relatively simple to implement compared to the standard DP-SGD implementations.

**Weaknesses:**

1. Algorithm 3 seems to be important, adding it to the paper would be nice.
2. There is an inherent tradeoff between privacy and utility, however the results seem to be the opposite, as in even for extremely small values of epsilon, the accuracy of model is pretty high, can the authors please explain this observation?
3. How would the results change if the models were trained from scratch instead of fine-tuning?

**Questions:**

How can the proposed method be extended to vision tasks? or vision generative models?
For instance NAR generative models solve a bert like optimization problem during training, is there a trivial way to extend this method?

---

> ### Author Response · Authors · 2023-11-17
> **Rebuttal by Authors**
>
> Thanks a lot for the positive feedback and constructive comments. Please find our response to each comment as below.
>
> > Algorithm 3 seems to be important, adding it to the paper would be nice.
>
> We will move Algorithm 3 to the main text in the new version. Thanks for the suggestion.
>
> > There is an inherent tradeoff between privacy and utility, however the results seem to be the opposite, as in even for extremely small values of epsilon, the accuracy of model is pretty high, can the authors please explain this observation?
>
> The privacy/utility tradeoff also exists in our results. Due to the benefit of our mechanism (small noise in a small privacy budget), the accuracy is already very high on a small privacy budget. Thus, sometimes the accuracy improvement is not observed to be significant (but accuracy is still increasing) while increasing the privacy budget. For instance, please see the privacy/utility trade offs in our results in Figure 3, Figure 4 and Table 3.
>
> >How would the results change if the models were trained from scratch instead of fine-tuning?
>
> Yes, similar to DP-SGD, LMO-DP can be applied to training from scratch. We did fine-tuning in this work since this is popular in practice for language models (LMs). It can also be applied to other applications, even in different domains (e.g., vision tasks). Due to the greatly reduced noise, we conjecture that similar performance will hold in training or other learning tasks.
>
> > How can the proposed method be extended to vision tasks? or vision generative models? For instance NAR generative models solve a bert like optimization problem during training, is there a trivial way to extend this method?
>
> Thanks for this question and the ideas for extension to other domains. It is very interesting to investigate new DP solutions (with better privacy/utility trade offs) for those tasks. For the traditional vision tasks, there may exist a trivial way to extend LMO-DP with the support of DP-SGD (though the LM Geometry should be revised since it is mainly designed for NLP tasks). More specifically, our proposed method can be extended to these vision tasks since the key component of our method is to search the less noise under the privacy constraints. Then, we still can have a similar framework to search less noise. However, for different domains (e.g., vision task), we may need to change the combination of mixture distributions and the loss functions and so on.
>
> For the vision generative models, if DP works for their training/fine-tuning, the accuracy can be further improved by adapting this method with proper extensions. To our best knowledge, in the case of NAR generative models, we may also need to design a new DP framework (and then optimize the noise) for the training as it solves a BERT like optimization problem during training.

---

### Official Review · Reviewer_tr6o · 2023-11-01

**Soundness:** 3 good
**Presentation:** 1 poor
**Contribution:** 4 excellent
**Rating:** 5
**Confidence:** 4

**Summary:**

The paper introduces a new way to train language models with differential privacy guarantees. Rather than adding Gaussian noise as done by DP-SGD and to the best of my knowledge all improvements that build on it, the authors build on a result by Mohammady (2020) and add a mixture of noise of several distributions. They extend the Renyi DP accountant to handle non-Gaussian noise. Finally, they also present empirically results showing that they can beat the DP-SGD state of the art. The improvements are particularly striking in the low $\varepsilon$ regime.

**Strengths:**

- Strong empirical results. The paper presents results for good model utility with very strong privacy guarantees of $\varepsilon < 0.1$. This is a significant improvement over current SOTA.
- The authors provided a repo to reproduce the result.
- The method is simple but very effective. It should be fairly straightforward to implement this method in most DP training libraries such as opacus which will be of great value to the community.
- Comprehensive evaluation

**Weaknesses:**

The main weakness of this paper is the presentation. The manuscript reads like a rushed submission that requires more careful proof reading. For example:
- What does LMO stand for? It is introduced as "Language model-based optimal" but Appendix C says it's "Laplace Mixture of Outcomes" which has not been introduced at all.
- Many references to the abstract for essential content which makes it difficult to read the paper fluently.
- Inconsistent values throughout the paper e.g. for $\alpha$ equation (2) states integers 2, ..., 129 whereas algorithm 1 states 3, ..., 130

**Questions:**

- Can you explain what you mean by LM geometry in more detail? I found it hard follow the LM Geometry section? What are the specific characteristics? Is $\mathbb{P}_{i/o}$ simply the conditional probability $\mathbb{P}[o|i]$? This seems to be a conventional language model. I'm failing to understand what aspect of this definition is geometric.
- Why did the authors chose RDP accounting over tighter accountants e.g. PRV accountant? PLRV based accountant should be able to handle multiple different noise distributions? Would that provide an even better privacy utility trade off?
- Can you explain figure 1 in more detail and how it shows that the mixture of distributions covers the whole space?
- It would be interesting to learn more about the mixture of noise distribution? What type of noise is the main contribution?
- The authors often refer to the privacy curve $\delta(\varepsilon)$ as the PLRV, while they are equivalent they're not the same.

As stated above, I believe the main weakness of the paper is the overall presentation. It believe if the authors could improve the writeup the value to the community would significantly higher.

---

> ### Author Response · Authors · 2023-11-17
> **Rebuttal by Authors**
>
> Thanks a lot for the positive feedback and constructive comments. Please find our response to each comment as below.
>
> > Presentation/Typos/Inconsistencies
>
> Thanks for these comments. We have fixed the typos/inconsistencies and added more clarifications. The new submission/manuscript will be updated soon.
>
> > Can you explain what you mean by LM geometry in more detail? I found it hard follow the LM Geometry section? What are the specific characteristics? Is $P_{\frac{i}{o}}$ simply the conditional probability $P[o|i]$? This seems to be a conventional language model. I'm failing to understand what aspect of this definition is geometric.
>
> Thanks for this question. LM Geometry refers to the geometric structure underlying the Language Model (LM). Let's delve into its components:
>
> - Parameter Space $\Omega$: this represents the configuration space of the LM. For instance, in a word embedding function, $\Omega$ could include parameters defining word vectors.
>
> - Mapping Functions $\mathcal{F}$: these functions map model configurations within $\Omega$ to specific characteristics. For example, in a word embedding function $v(w)$, $\mathcal{F}$ will map words $w$ to vectors in $\mathbb{R}^d$. Thus, $\mathcal{F}$ captures the transformation from model configurations to observable features.
>
> - Probability Distribution $\mathbb{P}_{i/o}$: this probability distribution characterizes the probabilistic transition from an input token $i$ to an output word $o$. In a standard language model, it indeed aligns with the conditional probability $P[o|i]$. However, the emphasis on "geometry" lies in the broader interpretation. It encapsulates the entire distribution of transitions, offering a geometric view of how the model navigates from one token to another probabilistically.
>
> Specific Characteristics:
>
> The specific characteristics captured by $\mathcal{F}$ might include the spatial arrangement of word vectors or the distribution of model weights. These characteristics contribute to the overall geometry, allowing us to analyze and quantify how the LM operates probabilistically.
>
> We will add more clarifications to the new version and update it asap.
>
> > Why did the authors chose RDP accounting over tighter accountants e.g. PRV accountant? PLRV based accountant should be able to handle multiple different noise distributions? (for Gaussian) Would that provide an even better privacy utility trade off?
>
> Thanks for this question. LMO-DP can use other accountants (orthogonal to them) as long as they can handle multiple different noise distributions (e.g., PLRV based accountant). RDP accounting is chosen for two reasons: (1) it can handle different noise distributions and provide a tight accountant, and (2) our Theorem 4.1 can prove that optimizing over RDP is equivalent to optimizing cross-entropy loss. Although PLRV may be able to provide even better privacy/utility trade off, LMO-DP may need non-trivial extensions to integrate it for efficiently optimizing the noise. We will add a discussion for this.
>
> > Can you explain figure 1 in more detail and how it shows that the mixture of distributions covers the whole space?
>
> In probability theory, to our best knowledge, there is no universally accepted measure for quantifying the comprehensiveness of a subset of probability functions. Therefore, Algorithm 2 (Quantification of LMO Search Space) is employed to assess the comprehensiveness of this subset in comparison to a universally simulated space, introducing a novel quantification test. We utilize three distance metrics (both probabilistic and deterministic): KL divergence, $\ell_2$ and EMD metrics to measure the distance between these two spaces. Our results demonstrate that, for any given noise 'n', there exists an LMO noise that remains close to 'n'. Please note for the EMD metric, we experience a small divergence (scaled by $10^{-3}$) when the domain of noise increases which necessitates a smaller quantization rate to achieve even better results.
>
> > It would be interesting to learn more about the mixture of noise distribution? What type of noise is the main contribution?
>
> Thanks for the question. It is a Laplace-based two-fold noise (the first-fold is the Laplace distribution while the second-fold is a mixture distribution). Specifically, the "inverse" of "scale parameter" of the Laplace distribution (1/b) is subject to the linear combination of Gamma, exponential and uniform distributions. In addition, we have added an ablation study to exhibit the contribution of noises (please see the supplementary document).
>
> > Privacy curve \delta(\epsilon) vs the PLRV
>
> We will fix it. Thanks.
>
> > As stated above, I believe the main weakness of the paper is the overall presentation. It believe if the authors could improve the writeup the value to the community would significantly higher.
>
> Thanks for the suggestion and encouraging feedback. We are working on improving the presentation/clarity of the work. The new version will be updated soon.

---

> > ### Comment · Reviewer_tr6o · 2023-12-05
> >
> > Thank you for your response and updating the manuscript. However, I still remain with my initial assessment. While I think the empirical results are very promising the work is held back by its presentation. In particular sections 3.3 and 4 are hard to follow and imprecise.

---

### Official Review · Reviewer_LUi6 · 2023-11-06

**Soundness:** 2 fair
**Presentation:** 2 fair
**Contribution:** 2 fair
**Rating:** 3
**Confidence:** 4

**Summary:**

This paper proposes LMO-DP, a framework for fine-tuning language models with DP. This work optimizes (minimizes) the noise added for DP to the cross-entropy in LM model training in a way that ensures a bounded RDP. The authors propose a sub-optimal reduction to the optimal optimization problem. The search space of LMO geometries is shown to be comprehensive. Altogether, this enables training of high utility models that outperform prior work by a significant margin, especially at low epsilons.

**Strengths:**

The main strength of this work is the strong empirical results. This work achieves extremely impressive empirical results that outperform prior work at even an order of magnitude larger epsilon. For example, Table 3 shows that LMO-DP at epsilon of 0.16 outperforms all prior work at even epsilon of 1.4.

To achieve these results, this work proposes a novel approach to optimizing (minimizing) the amount of noise added while satisfying a finite RDP guarantee. This is achieved by building on the work of Mohammady et al (2020).

These results are highly timely and significant, as LLMs are seeing increasing prevalence in academia and adoption in industry.

(*) Finally, as both a question and a strength, this work seems independent of LMs and can potentially have larger impact. The main crux seems to be optimizing over the cross-entropy loss which is used more broadly than just LMs.

**Weaknesses:**

Though this work achieves impressive results, it is unclear how sound these results are, for several reasons.

**First, the results presented are often times quite unclear and imprecise. This manifests in a few ways.**

(A) The results rely heavily on the supplemental material, with no proofs or high-level descriptions of proofs provided in the main-text. This makes it difficult to follow the line of thought and verify if the high-level approach is sound.

(B) One of the key algorithms is Algorithm 3 which is required to understand section 4.3 but does not appear in the main-text.

(C) Some of the terminology is often imprecise: e.g., (1) "corresponding parameters, including \theta and others" and (2) "secondary optimization" on page 6. What are "others" and where is this "secondary optimization". Neither seem to appear in Algorithm 3 (nor the initial optimization mentioned just prior).

(D) This work compares several orthogonal methods to DP on different (often unrelated axes) in a rather confusing way. At the heart of this work is minimizing the amount of noise needed by DP. Yet, this work lists and compares with largely orthogonal work that explores memory reduction (ghost clipping) or parameter efficiency techniques (which improve memory/noise scale through the # of parameters) but  cna be used in conjunction with these methods. Why does Table 1 give low memory to this work? This seems like a false claim considering this work does not optimize memory but instead the noise standard deviation. Also, this work claims faster convergence but does not provide any theoretical guarantee to convergence, only some empirical exploration.

**Second, the results in some places appear unsound, or, require additional explanation.**

(E) The comparison of noise of LMO-DP to standard Gaussian noise for DP is unclear. The work claims "less noise" and relies on Figure 2 to compare this for several choices of epsilon. However, due to the scale, it is very vague what the actual difference is. The figure should use a log scale and also report useful statistics like the average reduction. Interpreting this figure, it looks as though the chosen noise is often 0 (or, vanishingly small). This would be an extremely significant improvement over DP that requires much more analysis to both understand how/where this is coming from and to ensure correctness. One key analysis would be to show what the noise looks like under the extremes (i.e., any single component), and, to for example, show how this noise changes as a function of the components.

(F) The empirical details are lacking for both reproducibility and understanding of results. How large are the dataset sizes? How many classes are there? These details significantly influence DP learning.

(G) Figure 5 is also extremely difficult to interpret. Looking at it, it looks like this model can be trained in <10 steps. This is quite surprising. How many steps does non-private training take?

(H) Figure 4 and several tables indicate that LMO-DP with a full order magnitude less privacy cost can outperform DP-SGD. This is an outstanding feat that is likely due to weakness E) above. Understanding how this manifests is crucial and is currently underexplored.

(I) Though I am not very familiar with the work of Mohammady et al. (2020), their work requires specifying both the utility metric and the query (to be protected by DP). Theorem 4.1 currently only shows the loss/utility metric (cross-entropy) but does not define the query. This makes it seem as though the query is acutally releasing the loss and not the gradients which would lead to privacy leakage. Please clarify.

**Questions:**

(*) in strengths.

(F) In weaknesses.

(I) In weaknesses

---

> ### Author Response · Authors · 2023-11-17
> **Rebuttal by Authors (* in Strength and Weakness (A)-(D))**
>
> Thanks a lot for the insightful and constructive comments. Please find our response to each comment as below.
>
> >  (*) Seems independent of LMs and can potentially have larger impact.
>
> Thanks for this point. Yes, its applicability can easily extend to other training or fine-tuning scenarios with various adaptations. This is primarily due to two key characteristics of LMO-DP. First, it leverages the cross-entropy utility metric, which finds broad application in tasks across Language Models, Computer Vision, and Audio domains. Second, the universality established in Theorem 4.1 highlights a suboptimal solution based solely on minimizing Renyi-DP and optimization without imposing constraints.
>
> Renyi-DP has also demonstrated its versatility through extensive applications in diverse fields, showcasing compelling adaptations for enhanced efficiency and accuracy. The noise introduced by LMO-DP can be seamlessly integrated with these approaches.
>
> > (A) No proofs or high-level description in the main text.
>
> Thanks for the suggestion. We have restructured the core methodology section (to be updated soon).
>
> > (B) Algorithm 3
>
> We will move it to the main text. Thanks.
>
> > (C) Terminology
>
> Our methodology involves a grid search across 8 parameters, encompassing Gamma, exponential, and uniform distributions along with their corresponding weights, denoted as $a_1$, $a_2$, and $a_3$. We have refined the description for improved readability. “Others” refers to other LMO parameters, including weights of different randomization PDFs and PDF parameters. “Secondary optimization” refers to the optimization inside the predicate for the initial optimization in Equation (2): where R\'enyi-DP privacy bound $\epsilon_\alpha$ is defined as $\epsilon(\delta) = \min_{\alpha > 1}...$. It seeks to minimize the discrepancy between $\epsilon_{LMO}(\delta)$ and $\epsilon$.
>
> > (D) Comparison with orthogonal methods.
>
> Thanks for this comment.  Yes, our LMO-DP framework is orthogonal to several existing works (we focus on minimizing the noise). For this reason, in most of our experiments, we primarily compare it with the Gaussian mechanism for fine-tuning language models on accuracy. Finally, we added a small group of experiments to compare LMO-DP with SOTA methods. Notice that, although those memory reduction (e.g., Ghost Clipping) and parameter efficiency methods improve the performance from different aspects, they are also commonly considered as SOTA methods. We can also consider such comparisons as showing the improvements on accuracy for such orthogonal methods.
>
> > Low memory in Table 1
>
> Also, as clarified in Section 1, given such orthogonal contributions, our LMO-DP framework can boost the performance of those methods (e.g., Ghost Clipping) and inherit the benefits of other works (e.g., memory reduction). Thus, in our LMO-DP implementation (shown in the source code), we have two modes, including the one integrated with the Ghost Clipping with low memory and the same accuracy (please see line 1 in lmo-dp/experiments/private-transformers/commands_task1.sh, we can choose ghost or default for “--clipping_mode”). That is why we also give low memory to this work.
>
> > Faster Convergence
>
> Yes, our faster convergence is based on empirical observation (we will add a note for empirical observation or remove that column in Table 1). The possible reason for such empirically faster convergence lies in the dynamics of LMO-DP noise. By introducing a significantly smaller perturbation, as illustrated in Fig 2, the incorporation of LMO-DP noise in DPSGD-based technologies intriguingly leads to faster convergence.
>
> Specifically, as verified by our implementation results, LMO-DP noise for strong privacy guarantees (e.g., privacy budget $\epsilon < 0.2$) yields outcomes equivalent to those obtained with Gaussian and Laplacian noise when applied with larger budgets ($\epsilon > 2$), thanks to its noise function. The state-of-the-art in DP versus Convergence, exemplified by JIT [ICLR 21], has demonstrated that DP negatively impacts the convergence rate, especially for smaller epsilon values, leading to considerably longer convergence times of the SOTA methods.
>
> Hence, LMO noise effectively addresses two challenges (accuracy and convergence) simultaneously, making it a solution that hits two birds with one stone.

---

> ### Author Response · Authors · 2023-11-17
> **Rebuttal by Authors (Weakness (E)-(I))**
>
> Thanks a lot for the insightful and constructive comments. Please find our response to Weaknesses (E)-(I) as below.
>
> > (E) Scale in Figure 2 and report useful statistics.
>
> Thanks very much for this suggestion. We changed it to the log scale and reported the average reduction with more data samples (please see Figure A.1 in the supplementary file). The results for noise comparison look more clear (**the average reduction rate is 95.13% for $\epsilon=0.3$, 92.19% for $\epsilon=0.7$, 87.71% for $\epsilon=2$, and 87.31% for $\epsilon=3$**).
>
> > (F) The empirical details are lacking for both reproducibility and understanding of results. How large are the dataset sizes? How many classes are there?...
>
> Thanks for this comment. The optimal weights  $\alpha_1$, $\alpha_2$ and $\alpha_3$ for mixture distributions to generate the noise are in the range [0.1, 0.3]. After that, we use the full datasets and all the classes in the original datasets for fine-tuning (same as SOTA). SST-2 has more than 60k+ samples in the training set; QNLI has more than 100k+ samples; MNLI and QQP contain more than 350k but less than 400k samples for each dataset. SST-2, QNLI, and QQP include two classes each; MNLI includes three classes. The hyperparameters setting: batch size 2048, 6 epochs and sampling rate 2048/|D|. We will clarify it in the paper and the readme file for the code.
>
> > (G) Figure 5 is also extremely difficult to interpret...it looks like this model can be trained in <10 steps...
>
> This shows the steps that each method needs to achieve the same accuracy. Specifically, we evaluate the steps that need to reach 48.51% accuracy and 41.83% accuracy for the BERT-base and RoBERTa-base on the MNLI-m dataset; the steps that need to reach 70.49% accuracy and 71.09% accuracy for the BERT-base and RoBERTa-base on the QQP dataset; the steps that need to reach 48.98% accuracy and 45.67% accuracy for the BERT-large and RoBERTa-large on the MNLI-m dataset; the steps that need to reach 63.18% accuracy and 67.84% accuracy for the BERT-large and RoBERTa-large on QQP dataset.
>
> **We cannot compare the steps to achieve the highest/converged accuracy since the baseline(s) cannot achieve such high accuracy (given the same small privacy budget $\epsilon$).** Thus, we found the minimum of the highest accuracy for all the tasks on a specific dataset using similar-parameter models as the reference; then we count the steps to reach this reference.
>
> For instance, LMO-DP needs 1146 steps on the MNLI-matched dataset and 192 steps on the SST-2 dataset (6 epochs) using the roberta-base model. The non-private training needs relatively less (but comparable) steps.
>
> > (H) Figure 4 and several tables indicate that LMO-DP with a full order magnitude less privacy cost can outperform DP-SGD. This is an outstanding feat...
>
> Thanks for the suggestion. We will clarify this relationship between noise value and the language model performance in the revised version. Specifically, we propose a suboptimal solution for the cross-entropy optimization based solely on minimizing DP accountant, which helps us reduce the noise for gradients. In our revised version, we redraw Figure 2 to better visualize the comparison between the LMO-DP noise and Gaussian noise. With significantly reduced noise (e.g., high reduction rate), LMO-DP only needs a very small privacy cost to achieve high accuracy compared to DP-SGD.
>
> > (I) Though I am not very familiar with the work of Mohammady et al. (2020), their work requires specifying both the utility metric and the query (to be protected by DP). Theorem 4.1 currently only shows the loss/utility metric (cross-entropy) but does not define the query. This makes it seem as though the query is acutally releasing the loss and not the gradients which would lead to privacy leakage. Please clarify.
>
> This is a good question. Yes, Mohammady et al. (2020), requires specifying the utility metric and the query. We define the cross-entropy loss as the utility metric, and the optimization for noise/randomization is formulated at the level of training/fine-tuning (by considering all the gradients with accounted DP leakage as the query). Note that the LMO space is defined based on the overall DP guarantee for the entire training/fine-tuning, which ensures the same DP guarantee as the DP-SGD on noisy gradients with the RDP accountant.  In other words, noise (injected to the gradients) is optimally derived for the training/fine-tuning while ensuring the same overall DP guarantee as DP-SGD.
>
> Different from Mohammady et al. (2020), we theoretically prove that optimizing the noise based on the cross-entropy loss/utility metric can be approximated to be equivalent to search over the LMO space (Theorem 4.1). Then, the optimization problem can be more efficiently solved. Note that optimizing the noise at the gradient level may also be feasible, but based on a different metric and different space for DP guarantee of the gradient, which may not be efficiently solved.

---

> > ### Comment · Reviewer_LUi6 · 2023-12-04
> > **Score remains unchanged.**
> >
> > Thank you to the authors for provided the detailed rebuttal.
> >
> > I have read through the rebuttal as well as the other reviews and decide to keep my original score.
> >
> > First, I will highlight improvements addressed in the rebuttal. Thank you for clarifying the orthogonality of the approach with prior work, improving the presentation, and clarifying the terminology.
> >
> > However, I keep my score the same for the following reasons.
> >
> > First,  this work should use better baselines. In particular, this work proposes a new method to reducing noise based on optimizing to a particular  geometry (that is actually much broader than LLMs). Instead, this work compares to works in the LLM space that use largely orthogonal approaches and can even be used in conjunction. Instead, given the algorithmic contribution of this work, the baselines should be comparing to other noise reduction schemes in DP ML.
> >
> > Second, giving "low memory" is still confusing. I agree this can use other works to reduce memory, but this work does nothing inherently to reduce memory. Claiming this seems to confuse and claim that this work studies/improves memory tradeoffs which it does not.
> >
> > Third, I agree that DP can impact convergence through the noise. However, the empirical treatment is lacking and should be supplemented with theoretical convergence guarantees perhaps based on the noisy sgd literature.
> >
> > Fourth, theorem 4.1, it's proof, and the clarification from the rebuttal do not make it clear how the query is the DP gradient release. If P_{I/o} represents the model outputs given the input, which seems to be the case given the text in the proof, then we desire some bound on the sensitivity of the gradients and that the gradients satisfy some DP guarantee. As it stands, the proof is not clearly laid out to show that the theorem is correct. Though it may be, it cannot be evaluated based on the current manuscript.

---

### Author Response · Authors · 2023-11-19
**Revisions for the Manuscript**

Dear Reviewers,

Thank you so much for your constructive comments again. We have carefully revised the manuscript by taking into account all your insightful comments. Specifically, we have made the following major changes to improve the presentation and clarity.

- Added clarifications to the comments/questions by all the reviewers.

- Replotted Figure 2 (in the original manuscript) using the logarithmic scale and more random samples (with the same privacy guarantees), as well as derived the average reduction rates.

- Added the ablation study to the mixture of three noise distributions (Gamma, Exponential, and Uniform) in LMO-DP.

- Added more notes/comments for the proposed algorithms and moved the Algorithm 3 (in the original manuscript) to the main body.

- Fixed the typos/inconsistencies and applied editorial changes to align with the format and length for ICLR.

We hope the clarifications and revisions have addressed all the concerns. If you have any other questions, please feel free to let us know at your convenience. We look forward to hearing from you soon.

Sincerely,

Authors

---

### Meta-Review · Area_Chair_XZjY · 2023-12-06

**Metareview:**

Summary: The Authors propose a framework called "Language Model-based Optimal Differential Privacy (LMO-DP)" that aims to enable a tight composition of sub-optimal DP mechanisms for accurately fine-tuning language models.

Strengths: The LMO-DP mechanism seems to help significantly at stronger privacy regimes.

Weaknesses: There seems to be many typos, which makes it hard for the readers to judge the correctness of the technical/theoretical claims (See comments by tr6o and 4F9v).

Even discarding or downweighing one reviewer's evaluation (based on the misunderstandings as the authors noted), the overall evaluation by rest of the reviewers is still lower than that of papers that are typically accepted at ICLR.

**Justification For Why Not Higher Score:**

Two of the reviewers mentioned due to many typos, it is hard to assess the correctness of the theoretical claims. Writing should be significantly improved before publication.

**Justification For Why Not Lower Score:**

NA

---

### Decision · Program_Chairs · 2024-01-16

Reject